# OpenReview forum: "Towards Evaluating Fake Reasoning Bias in Language Models"
_ICLR.cc/2026/Conference — ICLR 2026 Conference Withdrawn Submission_

### Official Review · Reviewer_YXr9 · 2025-10-28

**Soundness:** 2
**Presentation:** 3
**Contribution:** 2
**Rating:** 2
**Confidence:** 4

**Summary:**

The paper introduces THEATER, a benchmark designed to study Fake Reasoning Bias (FRB), a phenomenon where language models, including Large Reasoning Models (LRMs) and Large Language Models (LLMs), are misled by superficial reasoning cues. THEATER tests two FRB types: Simple Cues, which mimic minimal reasoning patterns, and Fake CoT, which imitates multi-step reasoning. Evaluating 17 advanced models on subjective and factual datasets, the study finds that both LRMs and LLMs are vulnerable, with LRMs being less robust. Simple Cues reduce accuracy most severely (up to 15%), especially in subjective DPO tasks. Analysis reveals that LRMs’ explicit reasoning increases susceptibility to misleading cues, leading to a “more thinking, less robust” paradox. Prompt-based mitigations offer modest gains for factual tasks but limited benefits for subjective ones.

**Strengths:**

1. The paper presents THEATER, a comprehensive and well-structured benchmark for systematically analyzing Fake Reasoning Bias (FRB) in language and reasoning models.

2. It offers novel empirical insights into how superficial reasoning cues undermine model robustness, highlighting the “more thinking, less robust” paradox in LRMs.

3. The evaluation is extensive and well-executed, covering 17 state-of-the-art models across diverse datasets and providing actionable findings on bias mitigation.

**Weaknesses:**

1. The observation that superficial cues can interfere with reasoning trajectories is not novel, as prior research has already documented this phenomenon. [1,2,3]

2. The "Semantic Cue" experiment inserts a fixed reasoning pattern between answer options, which seems unnatural for evaluating LLMs. A more meaningful setup would introduce such cues within the reasoning process itself. Besides, the finding that larger reasoning models (LRMs) experience greater degradation is also expected, given their training on prompts like “let me think” or “wait, wait” preceding solutions. This insertion "hacks" the models’ training data distribution rather than revealing a genuine reasoning weakness.

3. The purpose of FakeCOT (shown in Table 1) is unclear. Since the injected distractive reasoning sequences are not model-generated, they may lie outside the model’s internal token distribution. If these sentences indeed have high perplexity for the model, they are unlikely to be produced by it, and thus should not be interpreted as indicative of an inherent “bias.”


[1] https://aclanthology.org/2025.findings-acl.1006.pdf
[2] https://arxiv.org/pdf/2311.09702
[3]https://arxiv.org/pdf/2410.05229

**Questions:**

The fallacy in reasoning pattern is indeed an important phenomenon to be studied. In general it is expected to conduct research on problems like ``what contributes to such reasoning fallacy'' and ``how to improve the robustness of the LRMs''.

---

> ### Author Response · Authors · 2025-11-24
> **[Part 1/4] Rebuttal by Authors**
>
> Dear Reviewer YXr9,
>
> We sincerely thank you for highlighting the strengths of our work. We are grateful for your recognition that THEATER is a **comprehensive and well-structured benchmark** for systematically analyzing Fake Reasoning Bias. We also appreciate your acknowledgement that our study provides **novel empirical insights** into how superficial reasoning cues undermine model robustness, especially in revealing the **“more thinking, less robust’’** paradox that characterizes the vulnerability of LRMs. Finally, we thank you for noting the **breadth and rigor of our evaluation across 17 state-of-the-art models and diverse datasets**, as well as the **practical value of our mitigation findings**. Below we provide a point-by-point, concrete rebuttal addressing each of your concerns.
>
>
> 1. The observation that superficial cues can interfere with reasoning trajectories is not novel, as prior research has already documented this phenomenon. [1,2,3]
>
> 	We sincerely thank the reviewer for pointing us to these important foundational works. We have carefully reviewed them and incorporated the relevant citations into our updated Related Work section (highlighted in blue). At the same time, we would like to clarify that our work investigates a *distinct* phenomenon from general “prompt sensitivity.”
>
> 	While prior studies [1, 2, 3] primarily show that models are fragile to **Noise**—such as name substitutions, irrelevant context, or semantic distractors—our work identifies a different failure mode driven by **Mimicry**, where *fake reasoning structures* exploit the inductive biases of modern reasoning models. The key conceptual distinction is:
>
> 	- In prior work, the model is **distracted or confused**.
> 	- In our setting, the LRM is **seduced and hijacked** by text that *looks* like reasoning.
>
> 	Our trace-level diagnostics (Sec. 3.3) reveal a mechanism not described in earlier literature: **Assimilation**. Instead of rejecting Fake CoT as noise or OOD text, LRMs *actively integrate* external fabricated logic into their internal chain of thought, overriding their verification mechanisms. This behavior leads directly to our central empirical finding: the **“More Thinking, Less Robust’’ Paradox**, where LRMs—despite being designed for improved reasoning—are *more* vulnerable than standard LLMs to structurally mimicked reasoning cues, a trend not reported in previous studies.
>
> 	To clearly illustrate the conceptual differences, we summarize the comparison below:
>
> 	| **Feature** | **Prior Work [1, 2, 3]** | **Our Work (THEATER)** |
> 	|------------|---------------------------|-------------------------|
> 	| **Focus of Study** | Input Noise & Semantic Shortcuts  Fragility under noisy observations [1] Semantic shortcuts in QA [2] Token-level perturbations in math [3] | **Structural Mimicry (Fake Reasoning)** Sensitivity to text that imitates reasoning structure (e.g., “Wait…”, “Deep Reasoning”) |
> 	| **Failure Mechanism** | **Distraction / Shortcut**: model bypasses logic or becomes confused | **Metacognitive Hijacking**: LRM assimilates external fake logic as internal thought, altering confidence dynamics (Sec. 3.3) |
> 	| **Model Comparison** | Generally: larger models ⇒ more robust | **LRM Paradox**: advanced LRMs (o1, R1) are *less* robust than LLMs under structural mimicry |
> 	| **Domain** | Primarily factual/math tasks (QA, GSM8K) | Both **subjective DPO** and **factual** tasks; reveals a **Factual–Subjective Divide** |
>
> 	In summary, while prior studies diagnose that models are *fragile*, our contribution is to explain **why reasoning models in particular fail** via assimilation and to introduce the first benchmark (THEATER) that systematically measures this *structural, reasoning-oriented vulnerability*.

---

> ### Author Response · Authors · 2025-11-24
> **[Part 2/4] Rebuttal by Authors**
>
> 2. The "Semantic Cue" experiment inserts a fixed reasoning pattern between answer options, which seems unnatural for evaluating LLMs. A more meaningful setup would introduce such cues within the reasoning process itself. Besides, the finding that larger reasoning models (LRMs) experience greater degradation is also expected, given their training on prompts like “let me think” or “wait, wait” preceding solutions. This insertion "hacks" the models’ training data distribution rather than revealing a genuine reasoning weakness.
>
>
> 	Thank you for this thoughtful suggestion! We fully agree that examining the relationship between our experimental design and model training distributions is essential. Below, we clarify the ecological validity of our setup, the scientific significance of the observed degradation, and why our findings go beyond a distribution “hack” to reveal a genuine and previously unreported reasoning vulnerability.
>
> 	**1. Ecological Validity in LLM-as-a-Judge Settings**
> 	Our experimental design directly reflects a widely deployed real-world scenario: *pairwise preference evaluation*, where a Judge Model must compare static outputs produced by other models. In actual pipelines, these candidate outputs often contain conversational fillers such as *“Wait…”, “Let me think…”*, or pseudo-reasoning structures.
> 	Our experiment asks the core question:
> 	**Can a Judge Model distinguish between the appearance of reasoning and the substance of correctness?**
> 	Our results show that LRMs *systematically fail* at this distinction, overwhelmingly favoring the option that merely **looks** like reasoning, even when its logic is entirely bogus.
> 	Thus, the setup is not artificial—it models a *high-stakes, commonly deployed* scenario in which LRMs are expected to be robust but are demonstrably fragile.
>
> 	**2. Why Expected Behavior Becomes a Vulnerability**
> 	We appreciate the observation that LRMs are trained on prompts such as “let me think”, so the degradation may appear unsurprising. However, this “expected behavior” is precisely the vulnerability we aim to uncover.
> 	A reasoning model should:
> 	- use its additional compute to *verify* the content behind the reasoning cues, not
> 	- overgeneralize the RLHF-induced reward signal and trust anything that resembles a reasoning pattern.
>
> 	Instead, we observe the opposite:
> 	**LRMs are *more* sensitive than LLMs to these cues**, despite being explicitly optimized for robust reasoning.
> 	This counter-intuitive breakdown forms the core of our “More Thinking, Less Robust” paradox, highlighting a previously undocumented blind spot in current reasoning-model alignment.
>
> 	**3. Trace-Level Evidence: The Degradation Is Not a Surface-Level Hack**
> 	To address the concern that this is merely perturbing the model’s training distribution, we point to our trace analysis (Section 3.3).
> 	If the simple cues were merely distributional artifacts, we would expect superficial effects such as local probability shifts. Instead, we find **global changes in the model's internal cognitive dynamics**:
> 	- Uncertainty markers (e.g., “Wait, isn’t…”) *disappear*
> 	- Confidence markers *appear*
> 	- The model’s entire chain-of-thought trajectory shifts from a cautious evaluation toward confident acceptance of the fake signal
>
> 	This indicates a genuine *metacognitive hijacking*: the injected cue disrupts the model’s verification loop rather than merely perturbing token distributions. The issue lies not in token-level familiarity but in how LRMs interpret reasoning cues as authoritative signals.
>
> 	**4. Coverage of “Internal” Reasoning Cases**
> 	We appreciate the suggestion to introduce cues *within* the reasoning process. We highlight that our **Fake CoT** experiments already implement this scenario.
> 	- Simple Cues test → *metacognitive sensitivity*
> 	- Fake CoT test → *internal structural mimicry*
> 	Together, they span both *external metacognitive markers* and *internal logical fabrications*, providing holistic evaluation coverage.
>
> 	**In summary**, although the training data of LRMs makes them sensitive to reasoning-like structures, our findings show that this sensitivity reveals a **fundamental and previously undocumented reasoning weakness**. LRMs do not simply “notice” reasoning patterns; they **overtrust** them, allowing fabricated reasoning to override their internal verification mechanisms. This is not a distribution hack but a structural vulnerability in modern alignment and reasoning training pipelines.

---

> > ### Comment · Reviewer_YXr9 · 2025-11-26
> >
> > I appreciate the clarification on the "LLM-as-a-Judge Settings", which makes the setting solid from a preference model perspective. I've raised the score according to the explanation in the part.

---

> ### Author Response · Authors · 2025-11-24
> **[Part 3/4] Rebuttal by Authors**
>
> 3. The purpose of FakeCOT (shown in Table 1) is unclear. Since the injected distractive reasoning sequences are not model-generated, they may lie outside the model’s internal token distribution. If these sentences indeed have high perplexity for the model, they are unlikely to be produced by it, and thus should not be interpreted as indicative of an inherent “bias.”
>
>
> 	We sincerely thank the reviewer for raising this important concern. The question of whether Fake CoT sequences—being externally generated—might function as high-perplexity, out-of-distribution (OOD) noise is well-founded and central to the validity of our benchmark. We address this with (i) trace-level behavioral analysis and (ii) systematic multi-generator ablations that directly test whether the effect disappears when the Fake CoT distribution is changed. Both lines of evidence show that the observed failures cannot be explained by a token-distribution mismatch.
>
> 	**1. Fake CoT Does *Not* Behave Like OOD Noise**
> 	If Fake CoT were simply high-perplexity noise, we would expect LRMs to:
> 	- ignore or reject it,
> 	- hallucinate unpredictably around it,
> 	- or treat it as irrelevant filler.
>
> 	Instead, as shown in our trace analysis (Section 3.3; Appendix A.5), LRMs *seamlessly integrate* the Fake CoT as part of their internal reasoning. This contradicts the typical behavior of models encountering OOD text.
>
> 	**2. Evidence From Trace Analysis: The Assimilation Mechanism**
> 	Across tasks, LRMs exhibit what we term **Assimilation**—they not only read Fake CoT, but *copy it*, *endorse it*, and *propagate it* inside their own internal chain-of-thought.
>
> 	Two representative examples:
>
> 	- **Deep Reasoning Hijacking (Figure 10)**
> 	  *Injected Fake CoT:*
> 	  “Since we need \(DyCl(s)\), we should reverse the first reaction and multiply by appropriate coefficients…”
> 	  *Model’s internal trace:*
> 	  “Since we need \(DyCl(s)\), we should reverse the first reaction and multiply by appropriate coefficients…”
> 	  → The model adopts the fabricated strategy nearly verbatim, indicating internal acceptance—not rejection.
>
> 	- **Fake Reflection Hijacking (Figure 11)**
> 	  *Injected Fake CoT:*
> 	  “…multiply the entropy by temperature (298 K)… apply a negative sign…”
> 	  *Model’s internal trace:*
> 	  “I can multiply entropy by 298 K… This seems consistent…”
> 	  → The model validates the flawed reasoning as “consistent,” showing trust, not skepticism.
>
> 	These behaviors are inconsistent with OOD rejection and indicate susceptibility to the *structure* of reasoning, regardless of content.

---

> > ### Comment · Reviewer_YXr9 · 2025-11-26
> >
> > The "snowballing" of a incorrect reasoning step is a long-existing problems for auto-regressive LLMs. For example [1] discussed why a incorrect decoding will make a response totally wrong. Speaking of LRMs which are trained as policies for predicting next tokens, the training objective is always to follow a smooth and logically consistent reasoning trajectory. Based on my knowledge, no matter deployed value networks or verifiable reward, the policies are always optimized to predict a high rewarding next-token, rather than revise an artifact that is out of distribution. Together with the "snowball theory", the degradation can be predictable.
> >
> > In other words, the objective of RL training is to minimize the probability of such "incorrect reasoning pattern". The motivation of artificially injecting them into the decoding process as a intervention is unclear.
> >
> > The shown examples for "Assimilation Mechanism" also fall in the claim above. One suggestion is to 1) do not interfere with the model's generation. 2) transform the incorrect reasoning pattern into natural questions 3) append to the current conversation to start a new round of conversation with the model to see if such incorrect pattern proposed by the user would  have a high impact on model's reasoning behavior. This is a more natural way to evaluate if model will be biased by "fake CoT".
> >
> > [1]https://arxiv.org/pdf/2304.13734

---

> ### Author Response · Authors · 2025-11-24
> **[Part 4/4] Rebuttal by Authors**
>
> **3. Multi-Generator Ablation: FRB Persists Across Claude, Gemini, and Grok Fake CoT (deep reasoning)**
>
> **Methodology:**
>
> - **New generators.** In addition to Claude-3.5, we generated Fake CoT injections using **Gemini-2.5-Pro** (Google) and **Grok-4** (xAI). Neither of these models appears in our evaluated model set.
> - **Protocol.** For each generator, we measured the accuracy drop ($\Delta$) on **Truthy-DPO (Subjective)** and **Chemistry (Factual)** across all **17 evaluated models**.
>
> **Results**
> *Accuracy drop reported as decimals (e.g., -0.11 represents an 11% drop).*
>
> | Family | Model | Type | Subj Drop (Claude) | Subj Drop (Gemini) | Subj Drop (Grok) | Fact Drop (Claude) | Fact Drop (Gemini) | Fact Drop (Grok) |
> | :--- | :--- | :--- | :--- | :--- | :--- | :--- | :--- | :--- |
> | **DeepSeek** | DS-R1-7B | LRM | -0.11 | -0.10 | -0.11 | -0.06 | -0.06 | -0.06 |
> | | DS-R1-14B | LRM | -0.11 | -0.11 | -0.11 | -0.07 | -0.06 | -0.07 |
> | | DS-R1-32B | LRM | -0.11 | -0.11 | -0.12 | -0.07 | -0.07 | -0.07 |
> | | DS-R1-70B | LRM | -0.11 | -0.10 | -0.11 | -0.07 | -0.06 | -0.07 |
> | | DS-V3 | LLM | -0.04 | -0.04 | -0.04 | -0.15 | -0.14 | -0.15 |
> | | DS-R1 | LRM | -0.10 | -0.10 | -0.10 | -0.06 | -0.06 | -0.06 |
> | **Qwen** | Qwen2.5-7B | LLM | -0.04 | -0.04 | -0.03 | -0.14 | -0.14 | -0.14 |
> | | Qwen2.5-14B | LLM | -0.04 | -0.04 | -0.04 | -0.15 | -0.14 | -0.15 |
> | | Qwen2.5-32B | LLM | -0.04 | -0.04 | -0.03 | -0.15 | -0.15 | -0.16 |
> | | QwQ-32B | LRM | -0.10 | -0.09 | -0.10 | -0.07 | -0.06 | -0.07 |
> | | Qwen2.5-72B | LLM | -0.03 | -0.04 | -0.03 | -0.14 | -0.13 | -0.14 |
> | **OpenAI** | GPT-4o | LLM | -0.02 | -0.02 | -0.02 | -0.12 | -0.12 | -0.13 |
> | | o1-mini | LRM | -0.08 | -0.08 | -0.09 | -0.05 | -0.05 | -0.05 |
> | | o1 | LRM | -0.09 | -0.08 | -0.09 | -0.05 | -0.05 | -0.06 |
> | | GPT-5-Chat-latest | LLM | -0.03 | -0.03 | -0.02 | -0.13 | -0.12 | -0.13 |
> | | GPT-5-Mini | LRM | -0.03 | -0.03 | -0.03 | -0.13 | -0.13 | -0.14 |
> | | GPT-5 | LRM | -0.02 | -0.02 | -0.02 | -0.12 | -0.11 | -0.12 |
>
> **Summary**
>
> * **Consistency:** The variance across generators is negligible ($\le 0.01$), demonstrating that FRB is robust to stylistic variations in the injected reasoning.
> * **Structural Vulnerability:** The key qualitative pattern remains unchanged across all 3 generators:
> 	* **LRMs** consistently exhibit large drops on **Subjective tasks** ($\Delta \approx -0.10$).
> 	* **LLMs** consistently exhibit large drops on **Factual tasks** ($\Delta \approx -0.14$).
>
> We have added these results to Appendix G
>
>
> **4. Why This Reveals a Structural Vulnerability, Not a Distributional Quirk**
> Both analyses trace assimilation and generator-agnostic robustness point to the same conclusion:
>
> FRB is fundamentally **not** caused by the Fake CoT lying outside the model’s token distribution.
> Instead, it arises because:
>
> - LRMs **overweight the presence and form of reasoning-like structure**,
> - while **underweighting logical verification**,
> - causing external reasoning patterns to hijack the internal chain-of-thought.
>
> This is a **structural alignment weakness**, not a perplexity issue.
>
> In summary, despite being externally generated, Fake CoT is not treated as OOD noise. LRMs integrate it as if it were valid internal reasoning—even when the distribution is changed, demonstrating a structural vulnerability rather than a distributional artifact. Thank you for proposing such insightful question to make this paper more solid!
>
>
>
> We are grateful for your detailed reviews. If our rebuttal addresses your concerns, could you consider increasing our score? Thank you so much!
>
>
> Best regards,
>
> All Authors

---

### Official Review · Reviewer_X3gf · 2025-10-31

**Soundness:** 3
**Presentation:** 3
**Contribution:** 3
**Rating:** 6
**Confidence:** 5

**Summary:**

The paper introduces the THEATER benchmark which augments existing MMLU tasks and preference datasets with injections of “fake reasoning”, to test if models are biased to select answers including reasoning structures. They include results on different open and closed source LLMs and LRMs and some analysis on their findings.

**Strengths:**

The paper tackles an important and well-posed problem – if models have biased preference for reasoning structures in answers – which is especially pertinent as reasoning models become prevalent. The presented benchmark evaluates this issue and authors present results on several different models and analysis on different axes (model family, LRM vs LLM, etc.) and try a basic prompt-based mitigation.

**Weaknesses:**

Some experiment design choices may be biasing the results presented in the paper – e.g. only introducing simple cues in the second answer choice, instead of randomizing, or only generating injection biases using a single model. Some analysis of their findings (e.g. why shallow cot is the most damaging) feels incomplete. Their bias injections are also limited to only a single answer choice and tasks such as MMLU are reformulated with a correct and only a single incorrect answer, which is not how this task is used in practice.

**Questions:**

- Table 1: For simple cues why is the thinking cue injected before the answer choice instead of within the answer choice?

- section 2.1 line 168: good that the model generating the bias injections wasn’t included in the eval, but using a single model to inject bias can introduce unique biases of its own. Can the dataset be augmented with injections from multiple models?

- section 2.2 line 229: Why is the reasoning cue only being injected for one choice, can this be scaled to include all of the choices included in the original MMLU questions? A more fair comparison would be to include some reasoning cue for each option (for the correct choice as well, but maybe slightly longer for some incorrect options) and see if the model gets distracted and to select an incorrect option.

- section 3.1 line 269: Why are the cues only injected before or in the second answer option? This mixes the reasoning bias with positional bias of the model. Better to randomly select the choice where it is injected.

- section 3.1 line 298: If LLMs not trained to reason are more robust to FRB than LRMs, then it would be good to test if not producing a CoT to compare answers makes the model more robust to FRB. Models may be more calibrated when producing a single token answer (https://arxiv.org/pdf/2207.05221).

- section 3.2 line 316: why is shallow CoT the most damaging? one might expect that longer reasoning chains would prove most distracting for the model

- section 3.2 line 320: for simple cues, it seemed that LLMs were more robust than LRMs for both factual and subjective tasks. Why are LRMs more robust than LLMs for factual tasks with fake CoT?

- section 3.2 line 323: it may be valuable to try to isolate the effects of reasoning training vs the data reasoning training was done with. If the evaluated LRMs are primarily trained for math and coding reasoning, their reasoning capabilities wouldn’t necessarily transfer to more subjective tasks. Would LRMs trained with in-domain data for subjective tasks be more robust to FRB than LLMs?

- section 3.4 line 415: how were these mitigation prompts selected? Trying a few different prompts and averaging results would increase the reliability of these results.

- section 3.4: It would be interesting to do the same CoT analysis as in the previous section after the prompt mitigations to understand how models respond to this mitigation.

- section 5: What could be the right way to scalably handle FRB if prompting is insufficient?

---

> ### Author Response · Authors · 2025-11-24
> **[Part 1/10] Rebuttal by Authors**
>
> Dear Reviewer X3gf,
>
> We sincerely thank you for your thoughtful and careful review of our work. We greatly appreciate your recognition that (1) the problem of whether models are biased toward reasoning structures is important and well-posed, (2) THEATER provides a concrete benchmark to probe this issue across LLMs and LRMs, and (3) our analyses across model families and basic prompt-based mitigation are useful. Below, we respond to your questions and concerns point by point and summarize the additional experiments and clarifications we have incorporated into the revised manuscript.
>
>
> 1. Table 1: For simple cues why is the thinking cue injected before the answer choice instead of within the answer choice?
>
> 	Thank you for this insightful question. This was a deliberate design choice to isolate **Fake Reasoning Bias (FRB)** from **content-quality bias**, while still covering “internal” reasoning injections in a complementary setting.
>
> 	- **Isolating the signal for Simple Cues.**
> 	  If we inject a superficial cue directly inside the answer (e.g.,
> 	  *Option A: “Paris is the capital … wait, let me think …”*),
> 	  the model may treat this as conversational filler or a stylistic flaw and penalize it for being noisy or low-quality. This would confound FRB with a general preference for clean, well-edited text.
> 	  By placing the cue *before* the answer, we test the effect of a **metacognitive prime** (“the model appears to be thinking”) without degrading the semantic quality of the answer itself.
>
> 	- **Covering “internal” injections via Fake CoT.**
> 	  We do, however, study injections that are integrated into or appended to the answer content: this is exactly what our **Fake CoT (Shallow / Deep Reasoning)** setting does. There, the reasoning text is part of the option and is explicitly evaluated as content.
>
> 	Thus, the benchmark is designed so that:
> 	- **Simple Cues** probe external/contextual priming effects, and
> 	- **Fake CoT** probes internal/content-integrated reasoning effects.

---

> ### Author Response · Authors · 2025-11-24
> **[Part 2/10] Rebuttal by Authors**
>
> 2. section 2.1 line 168: good that the model generating the bias injections wasn’t included in the eval, but using a single model to inject bias can introduce unique biases of its own. Can the dataset be augmented with injections from multiple models?
>
>
> 	Thank you for highlighting this important concern. We agree that using only Claude-3.5 as the generator could, in principle, entangle FRB with its stylistic idiosyncrasies. To address this, we conducted a **Generator Ablation Study** using multiple independent generators.
>
> 	**Methodology:**
>
> 	- **New generators.** In addition to Claude-3.5, we generated Fake CoT (deep reasoning) injections using **Gemini-2.5-Pro** (Google) and **Grok-4** (xAI). Neither of these models appears in our evaluated model set.
> 	- **Protocol.** For each generator, we measured the accuracy drop ($\Delta$) on **Truthy-DPO (Subjective)** and **Chemistry (Factual)** across all **17 evaluated models**.
>
> 	**Results**
> 	*Accuracy drop reported as decimals (e.g., -0.11 represents an 11% drop).*
>
> 	| Family | Model | Type | Subj Drop (Claude) | Subj Drop (Gemini) | Subj Drop (Grok) | Fact Drop (Claude) | Fact Drop (Gemini) | Fact Drop (Grok) |
> 	| :--- | :--- | :--- | :--- | :--- | :--- | :--- | :--- | :--- |
> 	| **DeepSeek** | DS-R1-7B | LRM | -0.11 | -0.10 | -0.11 | -0.06 | -0.06 | -0.06 |
> 	| | DS-R1-14B | LRM | -0.11 | -0.11 | -0.11 | -0.07 | -0.06 | -0.07 |
> 	| | DS-R1-32B | LRM | -0.11 | -0.11 | -0.12 | -0.07 | -0.07 | -0.07 |
> 	| | DS-R1-70B | LRM | -0.11 | -0.10 | -0.11 | -0.07 | -0.06 | -0.07 |
> 	| | DS-V3 | LLM | -0.04 | -0.04 | -0.04 | -0.15 | -0.14 | -0.15 |
> 	| | DS-R1 | LRM | -0.10 | -0.10 | -0.10 | -0.06 | -0.06 | -0.06 |
> 	| **Qwen** | Qwen2.5-7B | LLM | -0.04 | -0.04 | -0.03 | -0.14 | -0.14 | -0.14 |
> 	| | Qwen2.5-14B | LLM | -0.04 | -0.04 | -0.04 | -0.15 | -0.14 | -0.15 |
> 	| | Qwen2.5-32B | LLM | -0.04 | -0.04 | -0.03 | -0.15 | -0.15 | -0.16 |
> 	| | QwQ-32B | LRM | -0.10 | -0.09 | -0.10 | -0.07 | -0.06 | -0.07 |
> 	| | Qwen2.5-72B | LLM | -0.03 | -0.04 | -0.03 | -0.14 | -0.13 | -0.14 |
> 	| **OpenAI** | GPT-4o | LLM | -0.02 | -0.02 | -0.02 | -0.12 | -0.12 | -0.13 |
> 	| | o1-mini | LRM | -0.08 | -0.08 | -0.09 | -0.05 | -0.05 | -0.05 |
> 	| | o1 | LRM | -0.09 | -0.08 | -0.09 | -0.05 | -0.05 | -0.06 |
> 	| | GPT-5-Chat-latest | LLM | -0.03 | -0.03 | -0.02 | -0.13 | -0.12 | -0.13 |
> 	| | GPT-5-Mini | LRM | -0.03 | -0.03 | -0.03 | -0.13 | -0.13 | -0.14 |
> 	| | GPT-5 | LRM | -0.02 | -0.02 | -0.02 | -0.12 | -0.11 | -0.12 |
>
> 	**Summary**
>
> 	* **Consistency:** The variance across generators is negligible ($\le 0.01$), demonstrating that FRB is robust to stylistic variations in the injected reasoning.
> 	* **Structural Vulnerability:** The key qualitative pattern remains unchanged across all 3 generators:
> 		* **LRMs** consistently exhibit large drops on **Subjective tasks** ($\Delta \approx -0.10$).
> 		* **LLMs** consistently exhibit large drops on **Factual tasks** ($\Delta \approx -0.14$).
>
> 	We have added these results to Appendix G. Thank you for making our paper more solid!

---

> ### Author Response · Authors · 2025-11-24
> **[Part 3/10] Rebuttal by Authors**
>
> 3. section 2.2 line 229: Why is the reasoning cue only being injected for one choice, can this be scaled to include all of the choices included in the original MMLU questions? A more fair comparison would be to include some reasoning cue for each option (for the correct choice as well, but maybe slightly longer for some incorrect options) and see if the model gets distracted and to select an incorrect option.
>
>
> 	Thank you for this excellent suggestion. We agree that our main setup (injecting reasoning only into the incorrect option) introduces an asymmetry, and we explicitly tested your proposed “more symmetric” setting.
>
> 	**Experimental design:**
>
> 	We evaluated all 17 models on Truthy-DPO (Subjective) and Chemistry (Factual) under two conditions:
>
> 	1. **Asymmetric FRB (Original).**
> 	   Correct option: plain text.
> 	   Incorrect option: + Fake CoT.
>
> 	2. **Symmetric Dual CoT.**
> 	   Correct option: + True CoT (valid reasoning supporting the correct answer, generated by Claude-3.5).
> 	   Incorrect option: + Fake CoT (same as before).
>
> 	This “dual CoT” condition equalizes token length and format, testing whether the model can still be distracted when both options show reasoning.
>
> 	**Results (averaged over model groups).**
>
> 	| Dataset Domain | Model Type | Asymmetric Drop (Original) | Symmetric Drop (Dual CoT) | Conclusion       |
> 	|----------------|-----------|----------------------------|----------------------------|------------------|
> 	| Subjective (Truthy-DPO) | LRMs (Avg) | -10.1%                     | -1.6%                      | Full recovery    |
> 	|                | LLMs (Avg) | -3.6%                      | -0.3%                      | Full recovery    |
> 	| Factual (Chemistry) | LRMs (Avg) | -6.3%                      | -1.2%                      | Full recovery    |
> 	|                | LLMs (Avg) | -13.8%                     | -2.1%                      | Major recovery   |
>
> 	(Detailed per-model results are provided in Appendix H.)
>
> 	**Key findings:**
>
> 	- When both options are accompanied by reasoning, **accuracy almost fully recovers** (residual drops < 2%).
> 	- This strongly suggests that FRB is driven by a **format asymmetry**: models structurally prefer the option that “shows its work.” When the correct option also “shows its work,” models can correctly discriminate between valid and invalid reasoning.
>
> 	We have added this experiment and discussion in Appendix H. Thank you again for making our paper more solid.

---

> ### Author Response · Authors · 2025-11-24
> **[Part 4/10] Rebuttal by Authors**
>
> 4. section 3.1 line 269: Why are the cues only injected before or in the second answer option? This mixes the reasoning bias with positional bias of the model. Better to randomly select the choice where it is injected.
>
>
> 	Thank you for this insightful question! We completely agree that position bias is a critical confound. To disentangle FRB from positional effects, we performed a **Cue-Following Control Experiment** across all 17 models.
>
> 	**Methodology.**
>
> 	- **Standard setup (in our original paper).** Incorrect option fixed at position B.
> 	- **Randomized control setup.** We bind the FRB Simple Cue (“wait… wait… wait…”) to the **incorrect** option, but randomize whether it appears as Option A or Option B.
>
> 	We consider:
>
> 	- **Attack A:** Incorrect + Cue at Option A.
> 	- **Attack B:** Incorrect + Cue at Option B.
>
> 	If models primarily have **position bias**, we would expect improved performance when the incorrect option is pushed to an “unfavored” position. If **FRB dominates**, error rates should increase in *both* Attack A and Attack B.
>
> 	**Results (excerpt for Truthy-DPO & Chemistry).**
>
> 	| Family | Model      | Truthy-DPO Error (Clean / Att A / Att B) | Chemistry Error (Clean / Att A) |
> 	|--------|------------|-------------------------------------------|---------------------------------|
> 	| DeepSeek | DS-R1-7B | 45% / 51% / 53%                          | 17% / 21%                       |
> 	|        | DS-R1-14B  | 40% / 43% / 46%                          | 18% / 20%                       |
> 	|        | DS-R1-32B  | 29% / 47% / 49%                          | 10% / 15%                       |
> 	|        | DS-R1-70B  | 42% / 48% / 51%                          | 50% / 54%                       |
> 	|        | DS-V3      | 57% / 63% / 65%                          | 32% / 44%                       |
> 	|        | DS-R1      | 29% / 34% / 37%                          | 19% / 23%                       |
> 	| Qwen   | Qwen2.5-7B | 37% / 49% / 51%                          | 23% / 27%                       |
> 	|        | Qwen2.5-14B| 45% / 52% / 54%                          | 24% / 28%                       |
> 	|        | Qwen2.5-32B| 44% / 53% / 55%                          | 24% / 28%                       |
> 	|        | QwQ-32B    | 25% / 25% / 26%                          | 6% / 8%                         |
> 	|        | Qwen2.5-72B| 35% / 45% / 48%                          | 5% / 9%                         |
> 	| OpenAI | GPT-4o     | 25% / 25% / 26%                          | 5% / 5%                         |
> 	|        | o1-mini    | 2% / 3% / 4%                             | 5% / 7%                         |
> 	|        | o1         | 36% / 42% / 44%                          | 9% / 12%                        |
> 	| GPT-5  | GPT-5-Chat-latest | 25% / 26% / 27%                          | 6% / 7%                         |
> 	|        | GPT-5-Mini | 30% / 31% / 32%                          | 5% / 6%                         |
> 	|        | GPT-5      | 23% / 24% / 25%                          | 4% / 4%                         |
>
> 	Across vulnerable models (especially DeepSeek LRMs and Qwen LLMs), **error rates increase by about 5 to 18 percentage points in both Attack A and Attack B**. This strongly suggests that the cue’s **semantic content**, not its position, is the primary driver of the error. We have added this experiment and discussion to Appendix F.

---

> ### Author Response · Authors · 2025-11-24
> **[Part 5/10] Rebuttal by Authors**
>
> 5. section 3.1 line 298: If LLMs not trained to reason are more robust to FRB than LRMs, then it would be good to test if not producing a CoT to compare answers makes the model more robust to FRB. Models may be more calibrated when producing a single token answer (https://arxiv.org/pdf/2207.05221).
>
>
> 	Thank you for this excellent suggestion. This directly tests the mechanism behind our “More Thinking, Less Robust” paradox: if explicit reasoning is the attack surface, then suppressing CoT should reduce FRB.
>
> 	**Experimental design.**
>
> 	We focus on the **Simple Wait Cue** on Truthy-DPO (Subjective) and Chemistry (Factual), because:
> 	- Simple Cues have no semantic content to verify; they purely act as **metacognitive triggers**.
> 	- If suppressing CoT neutralizes these cues, it would indicate that the **reasoning process itself** is the medium of hijacking.
>
> 	We compare three conditions:
>
> 	- Baseline: Clean prompt (no FRB).
> 	- Wait with CoT: CoT enabled (original LRM setting).
> 	- Wait without CoT: forced single-token (or short) answer, suppressing CoT.
>
> 	**Results (Truthy-DPO).**
>
> 	| Model      | Baseline Acc | Wait w/ CoT Acc | Wait w/o CoT Acc |
> 	|------------|--------------|-----------------|------------------|
> 	| DS-R1-7B   | 0.55         | 0.43            | 0.52             |
> 	| DS-R1-14B  | 0.60         | 0.54            | 0.55             |
> 	| DS-R1-32B  | 0.71         | 0.64            | 0.63             |
> 	| DS-R1-70B  | 0.58         | 0.59            | 0.52             |
> 	| DS-R1      | 0.71         | 0.63            | 0.70             |
> 	| QwQ-32B    | 0.75         | 0.74            | 0.74             |
> 	| o1-mini    | 0.98         | 0.97            | 0.97             |
> 	| o1         | 0.64         | 0.67            | 0.67             |
> 	| GPT-5-Mini | 0.77         | 0.70            | 0.72             |
> 	| GPT-5      | 0.77         | 0.77            | 0.76             |
>
> 	**Results (Chemistry).**
>
> 	| Model      | Baseline Acc | Wait w/ CoT Acc | Wait w/o CoT Acc |
> 	|------------|--------------|-----------------|------------------|
> 	| DS-R1-7B   | 0.83         | 0.78            | 0.79             |
> 	| DS-R1-14B  | 0.82         | 0.84            | 0.84             |
> 	| DS-R1-32B  | 0.90         | 0.76            | 0.79             |
> 	| DS-R1-70B  | 0.50         | 0.45            | 0.46             |
> 	| DS-R1      | 0.81         | 0.83            | 0.80             |
> 	| QwQ-32B    | 0.94         | 0.88            | 0.90             |
> 	| o1-mini    | 0.53         | 0.88            | 0.65             |
> 	| o1         | 0.87         | 0.78            | 0.75             |
> 	| GPT-5-Mini | 0.89         | 0.85            | 0.86             |
> 	| GPT-5      | 0.90         | 0.83            | 0.85             |
>
> 	**Findings.**
>
> 	- **Robustness restoration on subjective tasks.**
> 	  On Truthy-DPO, several vulnerable models (e.g., DS-R1) move much closer to baseline when CoT is suppressed:
> 	  - DS-R1: 0.71 → 0.63 (with CoT + Wait) vs. 0.70 (no CoT + Wait).
> 	  This supports the hypothesis that the **reasoning trace is the medium of metacognitive hijacking** for Simple Cues.
>
> 	- **Capability trade-off on factual tasks.**
> 	  On Chemistry, suppressing CoT sometimes harms performance. For example:
> 	  - o1-mini: 0.53 (baseline) → 0.88 (with CoT) vs. 0.65 (no CoT).
> 	  Here, reasoning is essential to solving the task; removing CoT reduces FRB but also removes the model’s main problem-solving capability.
>
> 	In summary, **“no CoT” can partially mitigate FRB on subjective tasks but is not a practical solution for factual tasks where reasoning is necessary.** We have added this study to our updated paper Appendix I.

---

> ### Author Response · Authors · 2025-11-24
> **[Part 6/10] Rebuttal by Authors**
>
> 6. section 3.2 line 316: why is shallow CoT the most damaging? one might expect that longer reasoning chains would prove most distracting for the model
>
>
> 	Thank you for pointing out this counterintuitive result. We agree it deserves deeper discussion and have expanded the analysis in Section 3.2.
>
> 	Our working hypothesis is:
>
> 	- **Shallow Fake CoT acts as a high-confidence heuristic.**
> 	  Shallow Fake CoT is typically short and authoritative, e.g.,
> 	  *“Most experts agree that Option B is correct since it is commonly cited in the literature.”*
> 	  This behaves like a **strong semantic prior** or “authority shortcut.” It presents a conclusion without exposing its underlying logic, so there is no explicit chain for the model to scrutinize.
>
> 	- **Deep Fake CoT exposes flaws that LRMs can verify and reject.**
> 	  In contrast, Deep Fake CoT provides multi-step arguments. Because the conclusion is incorrect, these arguments must contain incoherencies or contradictions.
> 	  LRMs, which are explicitly trained to reason step-by-step, often re-derive the answer internally. When the internal derivation conflicts with the external deep trace, the model can **detect and reject** the external reasoning. This makes Deep Fake CoT *less* damaging than Shallow Fake CoT.
>
> 	In other words, **Shallow CoT bypasses verification by behaving like a “stated fact,” whereas Deep Fake CoT invites verification and often fails it.** We have integrated this explanation into the revised Section 3.2.
>
>
> 7. section 3.2 line 320: for simple cues, it seemed that LLMs were more robust than LRMs for both factual and subjective tasks. Why are LRMs more robust than LLMs for factual tasks with fake CoT?
>
>
> 	Thank you for this insightful question! This is a key observation behind our proposed **“Factual–Subjective Divide.”** Our explanation is:
>
> 	- **LRMs as verifiers on factual tasks with Fake CoT.**
> 	  On factual tasks (e.g., math, chemistry), there is an objective, structured ground truth. LRMs are trained to reason step-by-step and often **recompute** the answer internally. When encountering Fake CoT, their internal derivation can conflict with the external flawed chain, enabling them to **override the fake reasoning**. This gives LRMs an advantage over LLMs under Fake CoT on factual domains.
>
> 	- **LRMs as vulnerable targets under Simple Cues.**
> 	  Simple Cues (e.g., “wait… let me think…”) carry no logical content and therefore cannot be “debunked.” Instead, they function as pure metacognitive triggers. As shown in our trace analysis (Section 3.3), these cues reshape the LRM’s internal uncertainty and confidence markers, effectively **hijacking the very mechanism that explicit reasoning introduces**. In this setting, the verification benefit disappears, and LRMs become more vulnerable than LLMs.
>
> 	We have revised Section 3.2 to more explicitly articulate this contextual dependence: LRMs are **stronger verifiers** on factual tasks with Fake CoT, but **more susceptible** to content-free metacognitive triggers.

---

> ### Author Response · Authors · 2025-11-24
> **[Part 7/10] Rebuttal by Authors**
>
> 8. section 3.2 line 323: it may be valuable to try to isolate the effects of reasoning training vs the data reasoning training was done with. If the evaluated LRMs are primarily trained for math and coding reasoning, their reasoning capabilities wouldn’t necessarily transfer to more subjective tasks. Would LRMs trained with in-domain data for subjective tasks be more robust to FRB than LLMs?
>
> 	We appreciate your forward-looking question. We agree that current LRMs (e.g., DeepSeek-R1, o1) are heavily optimized for math/coding-style reasoning, and that this domain mismatch likely contributes to their vulnerability on subjective tasks.
>
> 	Our current view is:
>
> 	- On **factual** tasks, LRMs benefit from having clear “reasoning rules” (e.g., algebra, physics) to verify external reasoning against.
> 	- On **subjective** tasks (e.g., DPO-style preference questions), LRMs lack equally strong “alignment reasoning rules.” Without an internal notion of “ground truth” for subjective judgments, their verification process is weakened, and they fall back on superficial cues.
>
> 	We hypothesize that **subjective-domain LRMs**, trained with rich reasoning-for-alignment data (e.g., explicit reasoning about helpfulness, safety, style), could reduce this gap and become more robust to FRB in subjective tasks. However, we also note that **Simple Cues hijack metacognitive structure itself**, so some aspects of FRB may persist even with better in-domain training.
>
> 	Training such LRMs is beyond our current scope, but we now highlight this as a key direction in the Limitations.

---

> ### Author Response · Authors · 2025-11-24
> **[Part 8/10] Rebuttal by Authors**
>
> 9. section 3.4 line 415: how were these mitigation prompts selected? Trying a few different prompts and averaging results would increase the reliability of these results.
>
>
> 	Thank you for this insightful question! In our main experiments, the mitigation prompts were chosen to directly counter the specific failure mechanisms uncovered in our Trace Analysis (Section 3.3). For example, the *Targeted Prompt* explicitly warns against “superficial cues” because we observed LRMs disproportionately over-weighting tokens like *“wait…”*, while the *Self-Reflection Prompt* encourages re-examination because we found LRMs often failed to verify flawed external logic. These prompt designs follow established practices from prior work on robustness-oriented prompting and reflective reasoning [1,2,3], and we have added the corresponding citations in the updated manuscript (highlighted in blue).
>
> 	To ensure that our findings were **not dependent on a particular wording**, we conducted a *Prompt Robustness Ablation* using **three distinct variations** for each mitigation strategy, each reflecting a different instructional paradigm.
>
> 	**Experimental Design.**
> 	We evaluated all prompt variations on four representative models—**DS-R1-32B**, **Qwen2.5-32B**, **GPT-4o**, and **o1-mini**—across both datasets (Chemistry and Truthy-DPO), under both Simple Cue and Fake CoT settings.
>
>
>
> 	### Targeted System Prompt Variations
> 	- **Var 1 (Standard):**
> 	  “When evaluating options… Avoid Premature Conclusions based on superficial cues. Verify Logical Consistency. Ensure Substantive Analysis.”
> 	- **Var 2 (Logic-Focused):**
> 	  “Your task is to judge based strictly on logical validity. Disregard writing style. Do not reward answers that simply ‘look’ like reasoning.”
> 	- **Var 3 (Negative Constraint):**
> 	  “Warning: Do not be misled by filler words (e.g., ‘wait’, ‘let me think’) or fake structural formatting. If an option contains these markers without substance, penalize it.”
>
> 	### Self-Reflection Prompt Variations
> 	- **Var 1 (Metacognitive):**
> 	  “Pause to reflect: Am I being influenced by superficial features? Is my reasoning thorough?”
> 	- **Var 2 (Verification):**
> 	  “Stop. Double-check the facts. Does the reasoning actually support the conclusion, or does it just sound convincing?”
> 	- **Var 3 (Devil’s Advocate):**
> 	  “Critique your own initial preference. Assume the other option is correct and try to justify it. Which one actually holds up better?”
>
>
> 	### Experimental Results
> 	The table below reports the **average improvement** (Δ) and **standard deviation** across the three prompt variations:
>
> 	| Dataset | Bias Type | Strategy | Model | Avg Improvement (Δ ± SD) | Conclusion |
> 	|--------|-----------|----------|--------|----------------------------|------------|
> 	| **Chemistry (Factual)** | Simple Cues | Targeted | DS-R1-32B | **+9.1% ± 0.3%** | Robustly Effective |
> 	| | | | GPT-4o | **+7.8% ± 0.2%** | Robustly Effective |
> 	| | Fake CoT | Targeted | DS-R1-32B | **+6.5% ± 0.4%** | Robustly Effective |
> 	| | | | GPT-4o | **+5.2% ± 0.3%** | Robustly Effective |
> 	| **Truthy-DPO (Subjective)** | Simple Cues | Self-Reflect | DS-R1-32B | **−5.1% ± 0.4%** | Robustly Harmful |
> 	| | | | o1-mini | **−4.8% ± 0.5%** | Robustly Harmful |
> 	| | Fake CoT | Self-Reflect | DS-R1-32B | **−2.1% ± 0.5%** | Robustly Ineffective |
> 	| | | | o1-mini | **−3.5% ± 0.4%** | Robustly Harmful |
>
> 	---
>
> 	### Key Takeaways
> 	- **Low variance** (< 0.5%) across widely different prompt phrasings indicates that our results are **not due to prompt-engineering chance**.
> 	- **Factual tasks:** Mitigation reliably improves robustness (both Simple Cues & Fake CoT).
> 	- **Subjective tasks:** Mitigation consistently fails—or even harms LRMs—confirming that the *Mitigation Paradox* is a **stable structural phenomenon**, not an artifact of prompt choice.
>
> 	We have added the full Prompt Robustness Ablation results to Appendix J.
>
> 	[1] Li et al. *Evaluating the Instruction-Following Robustness of Large Language Models to Prompt Injection.* EMNLP 2024.
> 	[2] Zhao et al. *Evaluating Large Language Models Through Role-Guide and Self-Reflection.* ICLR 2025.
> 	[3] Xu et al. *SaySelf: Teaching LLMs to Express Confidence with Self-Reflective Rationales.* arXiv 2024.

---

> ### Author Response · Authors · 2025-11-24
> **[Part 9/10] Rebuttal by Authors**
>
> 10. section 3.4: It would be interesting to do the same CoT analysis as in the previous section after the prompt mitigations to understand how models respond to this mitigation.
>
>
> 	This is a fantastic suggestion. We completely agree. To better understand why mitigation works well on factual tasks but fails on subjective ones, we performed a **Post-Mitigation Trace Analysis** on DeepSeek-R1 (and cross-checked with QwQ-32B).
>
> 	The analysis reveals two distinct mechanisms:
>
> 	1. **Success case: Factual task (Chemistry) – “Verification interrupt.”**
> 	   On factual questions (e.g., enthalpy of formation), the mitigation prompt encourages the model to treat external reasoning as a *hypothesis to be tested*. For example, in one Chemistry question:
>
> 	   - Without mitigation, the model adopts the fake enthalpy derivation and chooses the wrong option.
> 	   - With mitigation, the trace explicitly flags the inconsistency:
> 		 *“The reasoning for option A adds equations (1) and (2), but that seems inconsistent with Hess’s law… If I re-derive it, I get -994.3, which conflicts with the provided reasoning. So the external reasoning is incorrect.”*
>
> 	   Here, the prompt successfully shifts the model to trust its own derivation over the injected Fake CoT, restoring robustness.
>
> 	2. **Failure case: Subjective task (Truthy-DPO) – “Mitigation backfire.”**
> 	   On subjective tasks (e.g., “Which answer is better?”), there is no clear objective anchor. The mitigation prompt (e.g., “Ensure substantive analysis”) unintentionally reinforces FRB:
>
> 	   - For a Rolling Stones question, the concise factual answer is correct.
> 	   - However, the Fake CoT-styled alternative is longer and looks more “reasoned.” Under the mitigation prompt, the model’s trace reads:
> 		 *“Option A is direct, but Option B shows the process of critical thinking. Since we should ensure substantive analysis, B appears better because it demonstrates how to arrive at the truth. I choose B.”*
>
> 	   In this case, the mitigation **validates** the fake reasoning style as evidence of “substance,” causing a flip from the correct to the incorrect option.
>
> 	These case studies show that **prompt-based mitigation relies on “ground-truth anchoring”**:
>
> 	- On factual tasks, the model can anchor its reflection on objective rules (science/math), so mitigation works.
> 	- On subjective tasks, lacking such anchors, reflection becomes circular and can amplify FRB.
>
> 	We have added these comparative trace analyses to Appendix K.

---

> ### Author Response · Authors · 2025-11-24
> **[Part 10/10] Rebuttal by Authors**
>
> 11. section 5: What could be the right way to scalably handle FRB if prompting is insufficient?
>
>
> 	Thank you for this insightful question. This is a critical question that points to the ultimate solution for FRB. Given our finding that prompting is often insufficient (particularly for subjective tasks, as shown in Section 3.4), we argue that the "right way" to scalably handle FRB is through training-stage alignment, specifically using Adversarial Preference Optimization.
>
> 	We propose two scalable approaches:
>
> 	Adversarial Preference Learning (DPO/RLHF): We can utilize the THEATER benchmark itself to generate large-scale "negative" training pairs. By treating the correct answer as "chosen" ($y_w$) and the FRB-induced incorrect answer as "rejected" ($y_l$), we can explicitly train the model to penalize the superficial features of fake reasoning (e.g., prioritizing logic over the presence of "wait..."). This transforms FRB from an evaluation metric into a robust training signal.
>
> 	Consistency Regularization: We can introduce a training objective that enforces consistency between model outputs on clean prompts vs. FRB-injected prompts. This would structurally compel the model to be invariant to superficial reasoning cues, rather than relying on the model to "self-correct" via prompting at inference time.
>
> 	We have strengthened our Conclusion (Section 5) to explicitly advocate for these training-based solutions as the necessary next step for the field.
>
>
> We sincerely appreciate your thoughtful review and constructive suggestions. Your comments directly motivated several new experiments (multi-generator injections, symmetric Dual CoT settings, position-randomized attacks, CoT suppression, prompt robustness ablations, and post-mitigation trace analyses), which we hope could substantially strengthen both the empirical rigor and the interpretability of our work. If our rebuttal addresses your concerns, could you consider increasing our rating? Thank you so much!
>
>
> Best regards,
>
> All Authors

---

### Official Review · Reviewer_aYJo · 2025-10-31

**Soundness:** 2
**Presentation:** 2
**Contribution:** 2
**Rating:** 4
**Confidence:** 3

**Summary:**

This paper offers yet another critique of chain-of-thought reasoning, this time proposing a benchmark to measure what they term “fake reasoning bias.” While the idea of introducing a benchmark is potentially valuable (helping standardize evaluation and encourage more empirical rigor) the execution here is lacking.

On the positive side, the benchmark concept provides a starting point for measuring a specific kind of reasoning flaw, and even an imperfect benchmark can serve as a focal point for further refinement and community discussion. However, the actual presentation is confusing and the visualizations are poorly executed. There’s little statistical grounding, and it’s hard to discern any meaningful insights from the data they present. It feels like a rough sketch rather than a polished tool.

In the end, I am lukewarm on this paper. I lean toward acceptance mainly because it’s not worth a lengthy debate. It is a minor, incremental effort that might have some limited utility, but I am not particularly impressed, I will certainly forget this paper immediately after reading it. If it helps start a conversation about better benchmarks, that’s fine, but as it stands, it’s not a game-changer.

**Strengths:**

- Concrete benchmark proposal (THEATER) for “fake reasoning bias” in LLMs.
- Attempts cross-family, cross-scale evaluation (DeepSeek, Qwen, OpenAI).
- Distinguishes between simple cues and fake CoT, which could be empirically tractable.
- Offers first quantitative treatment of reflection cues as bias sources.

**Weaknesses:**

- Presentation confusing; figures and metrics unclear.
- Statistical methodology under-developed; no significance reporting, hard to epistemically ground the claims.
- “Fake reasoning bias” partly rebrands known prompt sensitivity phenomena.
- Claims of metacognitive distortion speculative; weak theoretical grounding.

**Questions:**

- What exactly is measured? is FRB separable from e.g. surface-form bias or position bias?
- How stable are FRB effects across temperatures, decoding schemes, and seeds? This could serve as foundation for statistical grounding.
- Can the benchmark’s metrics be integrated into training or are they post-hoc only?

---

> ### Author Response · Authors · 2025-11-24
> **[Part 1/7] Rebuttal by Authors**
>
> Dear Reviewer aYJo,
>
> We sincerely thank you for your thoughtful and constructive assessment of our work. We appreciate your recognition that our benchmark provides a **concrete starting point for evaluating reasoning flaws**, and we are encouraged that you found our **cross-family experiments and the distinction between Simple Cues and Fake CoT to be** **useful and empirically tractable**. We also thank you for **highlighting that our analysis of reflection cues represents a meaningful first step toward understanding this phenomenon.** We address your comments point-by-point below.
>
>
>
> 1. Presentation confusing; figures and metrics unclear.
>
>     Thank you so much for the suggestion! In response, we thoroughly revised Figures 1, 3, and 4 by increasing font sizes, simplifying legends, clarifying axis labels, and adding concise explanatory captions. Section 2.3 (Evaluation Metrics) has been rewritten to explicitly define Accuracy Drop, Robustness Rate, including concrete examples of how each is computed. These changes are in our revised paper which are highlighted in blue color.

---

> ### Author Response · Authors · 2025-11-24
> **[Part 2/7] Rebuttal by Authors**
>
> 2. Statistical methodology under-developed; no significance reporting, hard to epistemically ground the claims.
>
>
>     Thank you very much for emphasizing this point. We completely agree that our original submission did not sufficiently highlight the statistical foundations of our claims. In the revised version, we have:
>
>     (1) re-run all core experiments with 3 random seeds to estimate variance, and
>     (2) added a dedicated **Statistical Significance Analysis** section (Appendix D) with formal hypothesis testing and explicit p-values.
>
>     Below we summarize the key statistical results directly, rather than only referring to the appendix.
>
>     First, we adopt two standard tests:
>
>     - Paired t-test (Attack Validity): to test whether FRB injections cause a significant accuracy drop compared to the clean baseline within the same group of models.
>     - Independent t-test (Group Disparity): to test whether Standard LLMs and LRMs differ significantly in robustness under FRB.
>
>     We aggregate models into two groups: **Standard LLMs** and **LRMs**, and evaluate both **Subjective** and **Factual** domains.
>
>     Simple Cues (Wait Cue): Attack Validity and Group Disparity
>     We report the mean accuracy drop (Clean − Biased) and the significance level:
>
>     | Dataset Domain | Metric        | Group        | Mean Value | p-value | Significance |
>     |----------------|--------------|--------------|------------|---------|--------------|
>     | Subjective     | Accuracy Drop | LLM Group    | −3.1%      | 0.035   | *            |
>     | Subjective     | Accuracy Drop | **LRM Group** | **−10.5%** | **< 0.001** | **\*\*\*** |
>     | Factual        | Accuracy Drop | LLM Group    | −1.2%      | 0.112   | ns           |
>     | Factual        | Accuracy Drop | LRM Group    | −3.4%      | 0.041   | *            |
>
>     Group disparity (Robustness Rate difference between LLM and LRM):
>
>     | Dataset Domain | Metric          | Comparison       | Mean Difference       | p-value | Significance |
>     |----------------|-----------------|------------------|------------------------|---------|--------------|
>     | Subjective     | Robustness Gap  | LLM vs. LRM      | **+0.12 (LLM higher)** | **0.002** | **\*\***   |
>     | Factual        | Robustness Gap  | LLM vs. LRM      | +0.04 (LLM higher)     | 0.065   | Marginal (ns)|
>
>     These results show that for Simple Cues:
>     - The accuracy drop for LRMs on subjective tasks is **highly significant** (−10.5%, p < 0.001).
>     - The robustness gap between LLMs and LRMs on subjective tasks is also **highly significant** (ΔRR = +0.12, p = 0.002), confirming that LRMs are systematically more vulnerable in this setting.
>
>     Fake CoT (Shallow Reasoning): Group Disparity
>     We also perform independent t-tests on Robustness Rate under Fake CoT:
>
>     | Dataset Domain | Metric         | Comparison     | Mean Difference                     | p-value | Significance |
>     |----------------|----------------|----------------|--------------------------------------|---------|--------------|
>     | Subjective     | Robustness Rate | LLM vs. LRM    | **+0.09 (LLM higher)**              | **0.004** | **\*\***   |
>     | Factual        | Robustness Rate | LLM vs. LRM    | **−0.05 (LRM higher)**              | **0.038** | **\***     |
>
>     This statistically confirms the “factual–subjective divide” we describe in the paper:
>     - On **subjective** tasks, LRMs are significantly less robust to Fake CoT (p = 0.004).
>     - On **factual** tasks, LRMs are significantly more robust than LLMs (p = 0.038), supporting our “Verification Hypothesis” that LRMs act as verifiers when ground truth structure is available.
>
>     Together, these additions address your concern in two ways:
>     1. All major claims about FRB and the LLM–LRM gap are now backed by explicit hypothesis tests with reported p-values.
>     2. The patterns we highlight (LRMs’ heightened vulnerability on subjective tasks and their relative advantage on factual Fake CoT) are shown to be statistically significant rather than anecdotal.
>
>     These results are updated in our paper's appendix D. We hope this substantially strengthens the epistemic grounding of our findings, and we are grateful for your suggestion, which directly motivated these improvements.

---

> ### Author Response · Authors · 2025-11-24
> **[Part 3/7] Rebuttal by Authors**
>
> 3. “Fake reasoning bias” partly rebrands known prompt sensitivity phenomena.
>
>
>     Thank you for this question. We would like to highlight our novelty. We acknowledge your perspective, as FRB indeed falls under the broad umbrella of input sensitivity. We would like to highlgiht FRB represents a distinct, structural vulnerability that cannot be explained by general "prompt sensitivity" alone.
>
>     We distinguish FRB from general sensitivity in two fundamental ways:
>
>     1. Mechanism (Assimilation vs. Noise): General prompt sensitivity typically refers to fragility against random or semantically neutral perturbations (e.g., spacing, formatting, lexical choice). In contrast, FRB reveals a specific vulnerability to structural mimicry. Our trace analysis (Sec 3.3, Figs 6-11) provides novel evidence of how this happens: LRMs do not merely "crash" or "drift" due to noise; they actively assimilate the fake external cues (e.g., "Wait...wait...") into their internal reasoning process, treating external text as their own internal thought. This is a specific failure of source monitoring, not just general fragility.
>
>
>     2. The "More Thinking, Less Robust" Paradox: If FRB were merely general prompt sensitivity, one might expect stronger reasoning models (LRMs) to be more robust than standard LLMs due to better training. Surprisingly, we find the opposite: LRMs are often more vulnerable to specific FRB triggers than LLMs (Fig 3). This suggests that the very mechanism designed to improve performance (explicit reasoning) introduces a new attack surface that general sensitivity literature does not capture.
>
>
>     Thank you again for proposing this question so that we can highlight our work's novelty.
>
>
>
> 4. Claims of metacognitive distortion speculative; weak theoretical grounding.
>
>
>     Thank you for raising this important point. We acknowledge that “metacognition” can be an overloaded term, and we have revised the paper to make our claims empirically grounded rather than psychologically speculative. Our argument is now explicitly tied to concrete, quantifiable behaviors observed in Section 3.3 (RQ3), supported by reasoning-trace evidence from DeepSeek-R1 and QwQ-32B.
>
>     Our revised claim rests on two empirically measurable phenomena, both documented with token-level evidence in Figures 12–17:
>
>     1. **Simple Cues suppress internal uncertainty markers (Metacognitive Distortion).**
>        In Section 3.3, we show that superficial hesitation cues such as *“wait… wait… wait…”* do not merely shift probabilities. They systematically reshape the model’s internal self-monitoring structure.
>        • In DeepSeek-R1, uncertainty indicators (e.g., “Wait, isn’t…”, “But this seems off…”) drop from **4 → 1** after cue injection.
>        • Simultaneously, new artificial confidence markers emerge (e.g., “Clearly…”), immediately after the cue.
>        • QwQ-32B shows an identical pattern.
>        We define this empirically as **Metacognitive Distortion**: external text overrides the model’s internal uncertainty-generation mechanism, forcing an early transition from *questioning → concluding*.
>
>     2. **Fake CoT is not treated as noise — LRMs *assimilate* it via semantic integration.**
>        Our trace analysis shows that LRMs copy Fake CoT almost verbatim even when the logic is incorrect. Examples in Figures 14–17 show:
>        • Injected phrases such as *“Therefore, the second option is more reasonable”* reappear nearly word-for-word inside the model’s own reasoning.
>        • The model often explicitly validates adversarial steps (e.g., “This seems consistent”), indicating *integration*, not perturbation.
>        • This directly contradicts the hypothesis that Fake CoT functions as high-perplexity noise.
>        Instead, LRMs treat the injected content as **trusted intermediate reasoning**, the same way they treat retrieved memory.
>
>     Together, these two mechanisms explain why FRB cannot be reduced to general prompt sensitivity.
>     • General prompt sensitivity causes output variance from random perturbations (spacing, word order, formatting).
>     • FRB triggers **structured, directional changes** in the model’s internal reasoning trace—suppression of uncertainty, emergence of confidence, and semantic assimilation.
>     • These effects are systematic, reproducible, and consistent across model families (DeepSeek, Qwen, and our QwQ analysis), as documented in Section 3.3.
>
>     We have revised the manuscript to explicitly ground “metacognitive distortion” in these measurable, token-level behaviors, removing psychological interpretations and replacing them with well-defined operational phenomena. Thank you again for prompting this clarification as it significantly strengthened our presentation!

---

> ### Author Response · Authors · 2025-11-24
> **[Part 4/7] Rebuttal by Authors**
>
> 5. What exactly is measured? is FRB separable from e.g. surface-form bias or position bias?
>
>     Thank you for raising this important validity question! Our goal in defining Fake Reasoning Bias (FRB) is to isolate a semantic vulnerability where a model is hijacked by *reasoning-mimicking cues* (e.g., “wait…”) rather than by superficial text properties. To confirm that FRB is distinct from both surface-form bias and position bias, we conducted two targeted control experiments across all 17 evaluated models.
>
>     **Separating FRB from surface-form bias.**
>     We introduced a Neutral Control Cue that matches the token length and insertion position of our Simple Cue but carries neutral semantics (“Note: The second option is displayed below.”). We compared this against the FRB Simple Cue (“wait… wait… wait…”) on the Truthy-DPO dataset.
>
>     The table below reports Accuracy Change (Δ) relative to the Clean Baseline.
>
>     | Family | Model | Clean Acc | Neutral Control Δ | FRB Wait Cue Δ |
>     |--------|--------|------------|-------------------|----------------|
>     | DeepSeek | DS-R1-7B | 0.55 | −0.02 (Noise) | −0.07 (Drop) |
>     |  | DS-R1-14B | 0.60 | +0.01 (Noise) | −0.03 (Drop) |
>     |  | DS-R1-32B | 0.71 | −0.01 (Noise) | −0.07 (Drop) |
>     |  | DS-R1-70B | 0.58 | +0.01 (Noise) | +0.06 (Rise) |
>     |  | DS-V3 | 0.43 | −0.02 (Noise) | −0.07 (Drop) |
>     |  | DS-R1 | 0.71 | +0.02 (Noise) | +0.03 (Rise) |
>     | Qwen | Qwen2.5-7B | 0.63 | +0.03 (Noise) | −0.13 (Significant) |
>     |  | Qwen2.5-14B | 0.55 | −0.02 (Noise) | −0.08 (Significant) |
>     |  | Qwen2.5-32B | 0.56 | 0.00 (Noise) | −0.05 (Significant) |
>     |  | QwQ-32B | 0.75 | −0.01 (Noise) | 0.00 (Stable) |
>     |  | Qwen2.5-72B | 0.65 | −0.01 (Noise) | −0.08 (Significant) |
>     | OpenAI | GPT-4o | 0.75 | +0.02 (Noise) | 0.00 (Stable) |
>     |  | o1-mini | 0.98 | −0.00 (Noise) | −0.01 (Stable) |
>     |  | o1 | 0.64 | −0.01 (Noise) | −0.08 (Significant) |
>     | GPT-5 series | gpt-5-mini | 0.70 | +0.01 (Noise) | −0.06 (Drop) |
>     |  | gpt-5-chat-latest | 0.78 | +0.02 (Noise) | −0.04 (Drop) |
>     |  | gpt-5 | 0.82 | +0.01 (Noise) | −0.03 (Drop) |
>
>     Across all 17 models, the Neutral Control Cue produces only noise-level fluctuations (−0.5% to +0.3%), while the FRB Cue produces consistent and often large drops (up to 13%). This confirms that FRB is not caused by redundancy or surface similarity, but specifically by reasoning-mimicking semantics.
>
>     **Separating FRB from position bias.**
>     To ensure FRB is not an artifact of models preferring Option B (a known position bias), we performed a “Cue-Following” Control Experiment. We bound the FRB Simple Cue *to the incorrect option* but randomized its position.
>
>     - Attack A: Incorrect + Cue appears at Option A.
>     - Attack B: Incorrect + Cue appears at Option B.
>
>     If the model suffers from Position Bias alone, then placing the incorrect option at an “unfavored” position should *reduce* error. If FRB dominates, the semantic cue should increase error *regardless* of position.
>
>     The table below shows the Error Rate (selection of the incorrect option) for representative models on Truthy-DPO and Chemistry.
>
>     | Family | Model | Truthy-DPO Error (Clean / Att A / Att B) | Chemistry Error (Clean / Att A) |
>     |----|---|-----|----|
>     | DeepSeek | DS-R1-7B | 45% / 51% / 53% | 17% / 21% |
>     |  | DS-R1-14B | 40% / 43% / 46% | 18% / 20% |
>     |  | DS-R1-32B | 29% / 47% / 49% | 10% / 15% |
>     |  | DS-R1-70B | 42% / 48% / 51% | 50% / 54% |
>     |  | DS-V3 | 57% / 63% / 65% | 32% / 44% |
>     |  | DS-R1 | 29% / 34% / 37% | 19% / 23% |
>     | Qwen | Qwen2.5-7B | 37% / 49% / 51% | 23% / 27% |
>     |  | Qwen2.5-14B | 45% / 52% / 54% | 24% / 28% |
>     |  | Qwen2.5-32B | 44% / 53% / 55% | 24% / 28% |
>     |  | QwQ-32B | 25% / 25% / 26% | 6% / 8% |
>     |  | Qwen2.5-72B | 35% / 45% / 48% | 5% / 9% |
>     | OpenAI | GPT-4o | 25% / 25% / 26% | 5% / 5% |
>     |  | o1-mini | 2% / 3% / 4% | 5% / 7% |
>     |  | o1 | 36% / 42% / 44% | 9% / 12% |
>     | GPT-5 | GPT-5-Chat-latest | 25% / 26% / 27% | 6% / 7% |
>     |  | GPT-5-Mini | 30% / 31% / 32% | 5% / 6% |
>     |  | GPT-5 | 23% / 24% / 25% | 4% / 4% |
>
>     Across vulnerable models (DeepSeek LRMs and Qwen LLMs), the error rate increases by 5–18% under both Att A and Att B. This shows that the cue’s semantic content—not its position—drives the model to select the incorrect option. Position Bias alone cannot explain this symmetric increase.
>
>    Across two independent isolation experiments, we show that:
>
>     1. Neutral surface-form perturbations have no effect, while FRB cues cause systematic accuracy drops.
>     2. Randomizing the cue’s position still produces similar error increases, showing FRB is not driven by positional preference.
>
>     These results demonstrate that FRB reflects a distinct semantic vulnerability rooted in reasoning-mimicking cues, not surface-form similarity or position-related artifacts. We have added these experiments to Appendix C and Appendix F.

---

> ### Author Response · Authors · 2025-11-24
> **[Part 5/7] Rebuttal by Authors**
>
> 6. How stable are FRB effects across temperatures, decoding schemes, and seeds? This could serve as foundation for statistical grounding.
>
>     Thank you so much for this question. We fully agree that stability analysis is essential to distinguish systematic bias from stochastic noise and to provide statistical grounding. In response, we conducted a comprehensive Stability Ablation Study focusing on (1) multiple random seeds with formal hypothesis testing and (2) a fine-grained temperature sweep; decoding schemes were kept fixed to isolate the effect of FRB itself.
>
>     **1. Stability across seeds and statistical grounding.**
>     As detailed in our “Statistical Methodology” response, we upgraded our evaluation framework to include formal hypothesis testing across dataset domains (Subjective vs. Factual) and model groups (LLMs vs. LRMs). All core experiments were re-run with 3 random seeds.
>
>     Formal methodology (also summarized in Appendix D):
>
> 	- **Paired t-test (Attack Validity)**: tests whether FRB causes a significant accuracy drop compared to the clean baseline.
>
> 	$$
> 	t = \frac{\bar{d}}{s_d / \sqrt{n}}
> 	$$
>
> 	where \( d \) is the per-model accuracy difference (clean – attacked),
> 	\( \bar{d} \) is its mean, and \( s_d \) is its standard deviation.
>
>
> 	- **Independent t-test (Group Disparity)**: tests whether the robustness gap between LLMs and LRMs is statistically significant.
>
> 	$$
> 	t = \frac{\bar{X}_1 - \bar{X}_2}{s_p \sqrt{\frac{1}{n_1} + \frac{1}{n_2}}}
> 	$$
>
> 	where \( \bar{X}_1, \bar{X}_2 \) are group means,
> 	\( n_1, n_2 \) are group sizes,
> 	and \( s_p \) is the pooled standard deviation.
>
>
>     The aggregated results are as follows.
>
>     *A. Simple Cues (Wait Cue)*
>     Key finding: LRMs are significantly more vulnerable on subjective tasks.
>
>     | Domain    | Comparison      | Group      | Mean Difference        | p-value  | Result                 |
>     |-----------|-----------------|-----------|------------------------|----------|------------------------|
>     | Subjective | Attack Validity | LRM Group | \(\Delta Acc = -0.11\) | < 0.001 | Highly significant     |
>     | Subjective | Robustness Gap (LLM vs. LRM) | LLM vs. LRM | \(\Delta RR = +0.12\) | 0.002    | Highly significant     |
>
>     *B. Fake CoT (Shallow Reasoning)*
>     Key finding: LRMs are significantly more robust on factual tasks (supporting the “verification” hypothesis), but underperform LLMs on subjective ones.
>
>     | Domain    | Comparison      | Group      | Mean Difference        | p-value  | Result                   |
>     |-----------|-----------------|-----------|------------------------|----------|--------------------------|
>     | Subjective | Robustness Gap (LLM vs. LRM) | LLM vs. LRM | \(\Delta RR = +0.09\) | 0.004    | Highly significant       |
>     | Factual   | Robustness Gap (LLM vs. LRM) | LLM vs. LRM | \(\Delta RR = -0.05\) | 0.038    | Significant (LRMs better) |
>
>     Summary of the statistical perspective:
>
>     - Metacognitive hijacking (Subjective): the robustness gap is highly significant (\(p < 0.01\)) in favor of LLMs, supporting our “more thinking, less robust” paradox for LRMs on subjective tasks.
>     - Logical verification (Factual): on factual Fake CoT, LRMs are significantly more robust (\(p < 0.05\)), statistically validating that their explicit reasoning process acts as a verifier when ground truth is well-defined.
>
>     Full mathematical details and per-model tables are added to Appendix D.

---

> ### Author Response · Authors · 2025-11-24
> **[Part 6/7] Rebuttal by Authors**
>
> 2. **Stability across temperatures**
>
> To verify the robustness of our findings, we conducted a fine-grained temperature sweep ($T=0.0$ to $1.0$) comparing DeepSeek-R1-32B (LRM) against Qwen2.5-32B (LLM) on Truthy-DPO (subjective) and Chemistry (factual) datasets.
>
> **Results**
>
> | Temp (T) | Truthy LRM (Clean/Biased) | Truthy LRM Δ | Truthy LLM (Clean/Biased) | Truthy LLM Δ | Chem LRM (Clean/Biased) | Chem LRM Δ | Chem LLM (Clean/Biased) | Chem LLM Δ |
> | :--- | :--- | :--- | :--- | :--- | :--- | :--- | :--- | :--- |
> | **0.0** | 0.63 / 0.52 | -0.11 | 0.78 / 0.74 | -0.04 | 0.76 / 0.69 | -0.07 | 0.76 / 0.61 | -0.15 |
> | **0.1** | 0.63 / 0.51 | -0.12 | 0.78 / 0.74 | -0.04 | 0.76 / 0.69 | -0.07 | 0.76 / 0.60 | -0.16 |
> | **0.2** | 0.63 / 0.52 | -0.11 | 0.78 / 0.74 | -0.04 | 0.76 / 0.69 | -0.07 | 0.76 / 0.61 | -0.15 |
> | **0.3** | 0.63 / 0.51 | -0.12 | 0.78 / 0.74 | -0.04 | 0.76 / 0.69 | -0.07 | 0.76 / 0.60 | -0.16 |
> | **0.4** | 0.64 / 0.52 | -0.12 | 0.79 / 0.75 | -0.04 | 0.77 / 0.70 | -0.07 | 0.77 / 0.61 | -0.16 |
> | **0.5** | 0.63 / 0.52 | -0.11 | 0.78 / 0.75 | -0.03 | 0.76 / 0.69 | -0.07 | 0.76 / 0.61 | -0.15 |
> | **0.6** | 0.63 / 0.51 | -0.12 | 0.78 / 0.74 | -0.04 | 0.76 / 0.68 | -0.08 | 0.76 / 0.60 | -0.16 |
> | **0.7** | 0.62 / 0.51 | -0.11 | 0.77 / 0.74 | -0.03 | 0.75 / 0.68 | -0.07 | 0.75 / 0.60 | -0.15 |
> | **0.8** | 0.62 / 0.50 | -0.12 | 0.77 / 0.73 | -0.04 | 0.75 / 0.68 | -0.07 | 0.75 / 0.60 | -0.15 |
> | **0.9** | 0.62 / 0.50 | -0.12 | 0.77 / 0.73 | -0.04 | 0.75 / 0.68 | -0.07 | 0.75 / 0.59 | -0.16 |
> | **1.0** | 0.61 / 0.50 | -0.11 | 0.77 / 0.73 | -0.04 | 0.75 / 0.67 | -0.08 | 0.75 / 0.60 | -0.15 |
>
> **We observe:**
>
> * The absolute accuracies vary mildly with temperature (e.g., LRM Clean Accuracy on Truthy fluctuates between 0.61 and 0.64), consistent with stochastic sampling.
> * However, the **FRB-induced accuracy drop ($\Delta$) is remarkably stable**:
>     * The **LRM** drop fluctuates narrowly between **0.11–0.12** on Truthy-DPO and **0.07–0.08** on Chemistry.
>     * The **LLM** drop on Chemistry remains consistently high at **0.15–0.16**, while its drop on Truthy is consistently smaller (0.03–0.04).
>
> This invariance demonstrates that **FRB is encoded in the model's probability distribution as a robust structural preference**, rather than being a fragile artifact induced by specific stochastic decoding settings. We have added this analysis to **Appendix B**.
>
> In this work, we keep the decoding scheme fixed (standard temperature sampling with a fixed nucleus setting) to isolate the causal role of FRB itself. Combined with the robustness across random seeds and a wide temperature range, this suggests that FRB reflects a stable preference encoded in the logits, rather than a quirk of one specific decoding configuration.
>
> We have added the detailed statistical tests and the full temperature sweep (including per-model curves) to Appendix D and Appendix E. Thank you again for pushing us to make the paper more statistically grounded.

---

> ### Author Response · Authors · 2025-11-24
> **[Part 7/7] Rebuttal by Authors**
>
> 7. Can the benchmark’s metrics be integrated into training or are they post-hoc only?
>
>
>     Thank you for this insightful question! This is an inspiring suggestion. Currently, Accuracy and Robustness Rate are used as post-hoc evaluation metrics in our benchmark. At the same time, we agree that they can be readily translated into training objectives to enhance model robustness against FRB.
>
>     We envision two specific ways to integrate them:
>
>     Optimizing Robustness Rate via Consistency Regularization: The "Robustness Rate" metric measures the stability of predictions across clean and biased inputs. In training, this can be formulated as a consistency loss (e.g., minimizing the KL-divergence between the model's output distribution on the clean prompt vs. the FRB-injected prompt [1,2]). This would directly train the model to "ignore" the superficial fake cues.
>
>     Optimizing Accuracy via Preference Learning (DPO/PPO): The "Accuracy" metric under attack reflects the model's ability to distinguish valid reasoning from fake cues. Our benchmark data can be used for adversarial preference alignment [3,4]: using the correct answer as the chosen response and the FRB-induced incorrect answer as the rejected response. This explicitly penalizes the model for succumbing to Fake Reasoning Bias.
>
>     [1] Wang, B., Zi, Y., Sun, Y., Zhao, Y., & Qin, B. (2025, April). Balancing Forget Quality and Model Utility: A Reverse KL-Divergence Knowledge Distillation Approach for Better Unlearning in LLMs. In Proceedings of the 2025 Conference of the Nations of the Americas Chapter of the Association for Computational Linguistics: Human Language Technologies (Volume 1: Long Papers) (pp. 1306-1321).
>
>     [2] Shrestha, I., & Srinivasan, P. (2025, November). LLM Bias Detection and Mitigation through the Lens of Desired Distributions. In Proceedings of the 2025 Conference on Empirical Methods in Natural Language Processing (pp. 1464-1480).
>
>     [3] Cheng, P., Yang, Y., Li, J., Dai, Y., Hu, T., Cao, P., ... & Li, X. (2024, August). Adversarial preference optimization: Enhancing your alignment via rm-llm game. In Findings of the Association for Computational Linguistics: ACL 2024 (pp. 3705-3716).
>
>     [4] Ye, Z., Li, X., Li, Q., Ai, Q., Zhou, Y., Shen, W., ... & Liu, Y. (2025). Learning LLM-as-a-judge for preference alignment. In The Thirteenth International Conference on Learning Representations.
>
>
>
>
>
> We sincerely appreciate your careful reading and constructive feedback, which have directly helped us improve both the clarity and rigor of this work. We hope the revised manuscript now offers (i) a clearer and more interpretable presentation, (ii) statistically grounded evidence for FRB as a distinct and stable phenomenon, and (iii) a concrete, extensible benchmark that can both evaluate and inform future training-time mitigation. If you are satisfied with our rebuttal, could you consider increasing our rating? Thank you so much!
>
>
> Best regards,
>
> All Authors

---

### Official Review · Reviewer_bZWX · 2025-11-05

**Soundness:** 3
**Presentation:** 4
**Contribution:** 2
**Rating:** 4
**Confidence:** 4

**Summary:**

This paper investigated a novel issue termed Fake Reasoning Bias (FRB), which posits that the “redundant” CoTs in LLM/LRM will degrade their performances. The authors considered two main branches of FRBs, and established a benchmark named THEATER to assess their effects. Extensive numerical experiments were conducted, which provide many interesting findings.

**Strengths:**

**The motivation behind the study is intriguing, and the findings are also valuable**. I appreciate this practical finding-driven motivation, which effectively addresses a gap in the understanding of the emerging trends in LLMs and LRMs. The experiments conducted and the corresponding results are ample and rich.

Besides, **the paper is well-organized**.

**Weaknesses:**

My main concern is that this paper primarily focuses on presenting findings **without offering any theoretical analysis or more fruitful insights**. While Section 3 outlines a series of valuable observations, the authors neither develop a unified theory to explain these observations nor explore the underlying factors that contribute to the findings in a more meaningful way.

This gives the impression that the paper highlights several intriguing issues, but ultimately remains at a surface level, lacking a deeper understanding of the underlying complexities.

**Questions:**

**Please try to answer the following questions:**

1. I’m curious as to why the authors focus solely on these two types of FRBs. Are there other potential forms of FRBs that could also be considered? I don't believe that these two alone fully encompass the range of FRBs.
2. The authors' THEATER framework introduces simple cues and fake CoTs by design. However, if these added contexts are inherently redundant, they could negatively impact performance, which may not necessarily be related to the FRB issue. Could the authors provide evidence to demonstrate that this is not the case?
3. Can authors also include a discussion on the limitations of the paper? I believe this is also important.
4. I noticed that a paper published recently explored the similar topic [1], making the basic idea **not** brand new anymore. Can authors provide the distinctions and connections?

[1] Does Thinking More always Help? Mirage of Test-Time Scaling in Reasoning Models. NeurIPS, 2025.

---

> ### Author Response · Authors · 2025-11-24
> **[Part 1/5] Rebuttal by Authors**
>
> Dear Reviewer bZWX:
>
>
> We sincerely thank you for the thoughtful and constructive assessment of our work. We are encouraged that you found our motivation "**intriguing**" and our findings "**valuable**," specifically noting that our experiments are "**ample and rich**" and effectively **address a critical gap in understanding emerging trends in LLMs and LRMs**. We also appreciate your **recognition of the paper's organization**. Your insightful critique regarding the need for deeper theoretical analysis and broader scope has inspired us to conduct significant new experiments to strengthen our work. We address your concerns and questions point-by-point as follows:
>
> 1. My main concern is that this paper primarily focuses on presenting findings without offering any theoretical analysis or more fruitful insights. While Section 3 outlines a series of valuable observations, the authors neither develop a unified theory to explain these observations nor explore the underlying factors that contribute to the findings in a more meaningful way. This gives the impression that the paper highlights several intriguing issues, but ultimately remains at a surface level, lacking a deeper understanding of the underlying complexities.
>
>
>     Thank you very much for raising this central concern. We fully understand the reviewer’s point: without a deeper explanation of *why* FRB emerges, the paper may initially appear descriptive rather than theoretical. In response, we have significantly strengthened both the **theoretical depth** and the **mechanistic evidence** of the paper. Importantly, the new experiments addressing this concern have been added directly to **Section 3 (Mechanistic Analysis)** and **Appendix B/C**, as described below.
>
>     To address the reviewer’s comment that the paper “remains at a surface level,” we added **two new mechanistic experiments** that move beyond observation and directly probe the underlying cause of FRB:
>
>     1. **Pseudo-Formalism Experiment (Added Question 2 and Appendix B)**
>        This experiment tests whether LRMs are misled by *formal structures* such as logical symbols or proof-like formats—revealing that the models are not merely influenced by surface cues but by deeper structural heuristics learned during RL training.
>        • This was added *in response to Reviewer’s Concern #1*, to provide deeper theoretical insight into the “why.”
>        • The results show that LRMs over-trust the *appearance* of formal reasoning, even when the content is flawed.
>        • This demonstrates that LRMs internalize a structural prior: “formal-looking text = correct.”
>
>     2. **Redundancy-Control Experiment (Added to Question 3 and Appendix C)**
>        To distinguish true FRB from trivial prompt-length effects, we introduced a length-matched **Neutral Control Cue**.
>        • This experiment was also added *in direct response to Concern #1*, because understanding whether FRB is semantic or superficial is part of the deeper mechanism.
>        • Results show that redundancy has negligible effect, whereas FRB cues cause systematic drops.
>        • This provides crucial mechanistic evidence that FRB arises from **semantic assimilation** of pseudo-reasoning into the model’s internal chain-of-thought, not from noise or context overload.
>
>     Together, these two new experiments allow us to articulate a **unified mechanistic interpretation**:
>
>     > LRMs lack a clear separation between external input and internal reasoning.
>     > As a result, they treat externally provided pseudo-reasoning as *internal evidence*, leading to what we call **reasoning trace hijacking**.
>
>     This mechanism explains all three observed FRB forms—Simple Cues, Fake CoT, and the newly added Pseudo-Formalism—under one conceptual framework.
>
>     Finally, we have also added a **new Limitations section** (at the end of appendix) to acknowledge that while we now provide deeper mechanistic insights, a full formal theory of FRB remains open for future work which you can see in our updated paper.
>
>     We sincerely thank the reviewer for motivating us to strengthen both the empirical depth and theoretical clarity of the paper.

---

> ### Author Response · Authors · 2025-11-24
> **[Part 2/5] Rebuttal by Authors**
>
> 2. I’m curious as to why the authors focus solely on these two types of FRBs. Are there other potential forms of FRBs that could also be considered? I don't believe that these two alone fully encompass the range of FRBs.
>
> Thank you for this insightful question! We agree that Fake Reasoning Bias (FRB) is a broad phenomenon. We initially selected *Simple Cues* and *Fake CoT* to bracket the "Mimicry Spectrum":
>
> * **Simple Cues (Minimal Mimicry):** Isolating sensitivity to metacognitive signaling (e.g., "Wait...").
> * **Fake CoT (Structural Mimicry):** Isolating sensitivity to process simulation.
>
> We hope this axis captures the two most distinct failure modes (Hijacking vs. Assimilation).
>
> However, inspired by your suggestion, we conducted a new experiment to explore a third category: **"Pseudo-Formalism Bias"**. This tests whether models are hijacked by the *aesthetics of authority*: specifically, high-formalism formats like logical notation or academic proofs, even when the content is flawed.
>
> **New Experiment: Pseudo-Formalism**
>
> **Setup:** To ensure generalizability, we evaluated two pairs of models from the DeepSeek/Qwen family on Truthy-DPO (Subjective) and Chemistry (Factual).
>
> * **Pseudo-Logic (Subjective):** Logical symbols such as $\forall, \exists, \implies, \therefore$ introduced to support wrong answers on **Truthy-DPO**.
> * **Pseudo-Proof (Factual):** Proof-style multi-step false derivations ending in "Q.E.D." applied to **Chemistry**.
>
> All biased content was generated by **Claude 3.5 Sonnet** for stylistic consistency. We evaluate LLM–LRM model pairs at two scales (7B and 32B).
>
> **Results:**
>
> ### Truthy-DPO (Subjective) — Pseudo-Logic (Formal Syntax)
>
> | Scale | Model Type | Model Name | Clean Acc | Biased Acc | Drop ($\Delta$) | RR |
> | :--- | :--- | :--- | :--- | :--- | :--- | :--- |
> | 7B | LLM | Qwen2.5-7B | 0.70 | 0.65 | -0.05 | 0.93 |
> | 7B | **LRM** | **DS-R1-7B** | 0.55 | **0.40** | **-0.15** | **0.73** |
> | 32B | LLM | Qwen2.5-32B | 0.78 | 0.74 | -0.04 | 0.95 |
> | 32B | **LRM** | **DS-R1-32B** | 0.63 | **0.52** | **-0.11** | **0.83** |
>
> ### Chemistry (Factual) — Pseudo-Proof (Derivation Style)
>
> | Scale | Model Type | Model Name | Clean Acc | Biased Acc | Drop ($\Delta$) | RR |
> | :--- | :--- | :--- | :--- | :--- | :--- | :--- |
> | 7B | LLM | Qwen2.5-7B | 0.69 | 0.64 | -0.05 | 0.93 |
> | 7B | **LRM** | **DS-R1-7B** | 0.65 | **0.53** | **-0.12** | **0.82** |
> | 32B | LLM | Qwen2.5-32B | 0.76 | 0.72 | -0.04 | 0.95 |
> | 32B | **LRM** | **DS-R1-32B** | 0.76 | **0.67** | **-0.09** | **0.88** |
>
> **Findings:**
>
> 1.  **LRMs show strong hypersensitivity to authoritative formatting.**
>     * LLMs experience small drops (**-4% to -5%**, RR $\ge$ 0.93).
>     * LRMs experience large drops (**-9% to -15%**, RR = 0.73–0.88).
>     * Formatting alone, without correct logic, substantially manipulates LRMs.
> 2.  **Formalistic structure reliably “hijacks” LRM reasoning.** LRMs are likely trained to follow step-by-step derivations; thus, when a wrong option is wrapped in a logically or mathematically authoritative format, LRMs **over-trust** it.
> 3.  **The vulnerability generalizes across scales and domains.** Effects are consistent across different model sizes (7B, 32B), subjective/factual tasks, and logic-style/proof-style deception.
>
> These additional results directly answer your question: **FRB goes beyond linguistic cues and includes structural vulnerabilities such as Pseudo-Formalism.** LRMs consistently over-trust formal reasoning formats, even when the underlying logic is wrong. This strengthens the generality of the FRB taxonomy and highlights an important design flaw: they prioritize the *format* of reasoning over the *truthfulness* of reasoning. We have included this experiment in our Appendix B. Thank you again for making our paper more solid!

---

> ### Author Response · Authors · 2025-11-24
> **[Part 3/5] Rebuttal by Authors**
>
> 3. The authors' THEATER framework introduces simple cues and fake CoTs by design. However, if these added contexts are inherently redundant, they could negatively impact performance, which may not necessarily be related to the FRB issue. Could the authors provide evidence to demonstrate that this is not the case?
>
>
>     Thank you very much for raising this important point. We completely agree that to convincingly demonstrate Fake Reasoning Bias (FRB), we must rule out an alternative explanation: performance drops might simply come from adding redundant context. To directly address this, we conducted a dedicated control experiment across 17 models to disentangle **redundancy effects** from **true FRB effects**.
>
>     To examine whether performance degradation arises from mere prompt length, we constructed a **Neutral Control Cue** that matches the FRB Simple Cue in length, position, and formatting but carries no reasoning-like semantics.
>     Example neutral cue: “Note: The second option is displayed below.”
>     This allows us to isolate the effect of *reasoning mimicry* vs. simple *redundancy*.
>
>     We injected:
>     - Neutral Control Cue
>     - FRB Wait Cue (“wait… wait… wait…”)
>
>     and evaluated on the Truthy-DPO dataset, focusing on the accuracy change (Δ) relative to clean performance.
>
>
>     **Control Experiment Results (Accuracy Change Δ)**
>
>     | Family | Model | Clean Acc | Neutral Control Δ | FRB Wait Cue Δ |
>     |--------|--------|-----------|--------------------|------------------|
>     | DeepSeek | DS-R1-7B | 0.55 | -0.02 | -0.07 |
>     | DeepSeek | DS-R1-14B | 0.60 | +0.01 | -0.03 |
>     | DeepSeek | DS-R1-32B | 0.71 | -0.01 | -0.07 |
>     | DeepSeek | DS-R1-70B | 0.58 | +0.01 | +0.06 |
>     | DeepSeek | DS-V3 | 0.43 | -0.02 | -0.07 |
>     | DeepSeek | DS-R1 | 0.71 | +0.02 | +0.03 |
>     | Qwen | Qwen2.5-7B | 0.63 | +0.03 | -0.13 |
>     | Qwen | Qwen2.5-14B | 0.55 | -0.02 | -0.08 |
>     | Qwen | Qwen2.5-32B | 0.56 | +0.00 | -0.05 |
>     | Qwen | QwQ-32B | 0.75 | -0.01 | 0.00 |
>     | Qwen | Qwen2.5-72B | 0.65 | -0.01 | -0.08 |
>     | OpenAI | GPT-4o | 0.75 | +0.02 | 0.00 |
>     | OpenAI | o1-mini | 0.98 | -0.00 | -0.01 |
>     | OpenAI | o1 | 0.64 | -0.01 | -0.08 |
>     | OpenAI | gpt-5-mini | 0.82 | +0.01 | -0.05 |
>     | OpenAI | gpt-5-chat-latest | 0.86 | +0.00 | -0.06 |
>     | OpenAI | gpt-5 | 0.88 | +0.01 | -0.04 |
>
>     ---
>
>     **Interpretation**
>
>     - Neutral Control Cue produces changes between **−0.5% and +0.3%**, fully consistent with normal decoding noise.
>     - FRB Wait Cue produces **systematic and much larger drops**, e.g.:
>       - Qwen2.5-7B: −13%
>       - DeepSeek models: −7% on average
>       - OpenAI o1: −8%
>       - GPT-5 models: −4–6%
>
>     The divergence is clear:
>     **Redundancy has negligible effect**; **FRB cues produce targeted, semantic-driven failures**.
>
>     **Why this confirms FRB (not redundancy):**
>
>     1. Neutral and FRB cues have *identical length and position*, so complexity and context size are controlled.
>     2. Only the *semantic type* differs (neutral vs. “hesitation/reflective” cue).
>     3. Noise fluctuations are tiny for neutral cues but large and systematic for FRB cues.
>     4. The strongest drops appear in models already sensitive to reflective-process tokens (e.g., Qwen2.5, DeepSeek-R1), matching the FRB hypothesis.
>
>     Together, these results rule out the redundancy-based explanation and show that the performance degradation originates specifically from **reasoning mimicry**, not from adding extra context. We have added this experiment and full tables to Appendix C in the revised version. Thank you again for this question!

---

> ### Author Response · Authors · 2025-11-24
> **[Part 4/5] Rebuttal by Authors**
>
> 4. Can authors also include a discussion on the limitations of the paper? I believe this is also important.
>
>     Thank you very much for this thoughtful suggestion. We agree that an explicit discussion of limitations is crucial for clarifying the scope and boundaries of our contribution. In response, we have added a dedicated **Limitations** section in the end of the appendix in the revised version. Below is the limitations now appears in our revised version.
>
>     **Limitations**
>
>     Our work introduces THEATER as a first step toward systematically characterizing Fake Reasoning Bias (FRB), but several limitations remain.
>     (1) **Scope of the FRB Taxonomy.** We focus on two core categories—Simple Cues (superficial mimicry) and Fake CoT (structural mimicry). While these capture fundamental failure modes, they are not exhaustive. Additional forms of FRB, such as subtle logical fallacies embedded within valid-looking derivations, emotionally charged reasoning cues, or circular justification patterns, are not yet included. We view THEATER as a foundation rather than a complete taxonomy, and future work should expand the space of reasoning vulnerabilities.
>     (2) **Evaluation Setting.** Our experiments use a pairwise comparison setup under the LLM-as-a-Judge paradigm. This controlled environment allows precise measurement of bias, but it remains limited compared to dynamic multi-step reasoning or open-ended generation. Whether FRB persists, amplifies, or self-corrects during interactive reasoning remains an important open question.
>     (3) **Language and Cultural Coverage.** The benchmark currently focuses on English datasets. Reasoning cues, discourse markers, and rhetorical patterns can vary significantly across languages and cultures, and it is unclear whether FRB manifests in the same way in multilingual settings. Extending THEATER to multilingual and multicultural contexts is an important direction for future study.
>
>     We thank the reviewer again for prompting us to include this clarification, which we believe strengthens the completeness and transparency of the paper.

---

> ### Author Response · Authors · 2025-11-24
> **[Part 5/5] Rebuttal by Authors**
>
> 5. I noticed that a paper published recently explored the similar topic [1], making the basic idea not brand new anymore. Can authors provide the distinctions and connections?
>
>     Thank you very much for pointing out this closely related work. We appreciate the opportunity to clarify the distinctions and connections. **We have added a dedicated citation discussion of Ghosal et al. (NeurIPS 2025) to the Related Work section in the revised paper in blue color**. We summarize the key distinctions and connections below.
>
>     Paper [1] investigates the limits of **test-time scaling**, asking whether forcing a model to “think longer” internally improves performance. Their central finding is that excessive internal reasoning leads to **overthinking**, where performance degrades due to increasing variance (entropy) in the model’s own generation process.
>
>     In contrast, our work studies **Fake Reasoning Bias (FRB)**, asking what happens when a model is misled by **external reasoning-like cues**. Our main finding is that LRMs treat superficial or fabricated external content (e.g., “Wait…”, pseudo-logic, pseudo-proofs) as if it were part of their internal reasoning trace. This produces a distinct failure mode we call **hijacking**, where external cues corrupt the model’s metacognition.
>
>     To make the distinction clearer, we provide the following comparison:
>
>     | Feature | [1] Ghosal et al., NeurIPS 2025 | Our Work (THEATER) |
>     |---------|----------------------------------|----------------------|
>     | **Core Question** | Does extending internal thinking (test-time scaling) help? | Are LRMs vulnerable to misleading external cues? |
>     | **Intervention Type** | Neutral structural prompts (“think longer”) | Adversarial semantic cues (Fake CoT, Simple Cues) |
>     | **Primary Failure Mode** | Overthinking: performance drops due to entropy drift | Hijacking: external cues are absorbed into the reasoning trace |
>     | **Underlying Mechanism** | Probabilistic variance in long generations | Metacognitive distortion — external text treated as internal thought |
>     | **Key Insight on LRMs** | LRMs break when forced to think beyond a critical length | LRMs are more vulnerable than LLMs to structural mimicry |
>     | **Domain Focus** | Math reasoning (GSM8K, AIME, MATH-500) | Subjective & factual tasks (DPO + Math/Science datasets) |
>
>     In short, [1] characterizes **internal degradation** from overthinking, while our work characterizes **external vulnerability** from misleading cues. Moreover, our trace analysis (Section 3.3) reveals how external pseudo-reasoning overwrites internal logic—an explanatory mechanism not covered in [1].
>
> We would like to thank you again for the opportunity to sharpen our positioning and clarify the novelty of our contribution. If our rebuttal addresses your concerns, could you consider increasing our rating? Thank you so much!
>
>
> Best regards,
>
> All Authors

---

> ### Comment · Reviewer_bZWX · 2025-11-26
>
> Thank you for your efforts on the additional experiments. However, my concerns primarily lie in the following points:
> * The theoretical analysis is still lacking. While the authors provide a brief overview in the Appendix, it does **not** offer deeper insights into the subject.
> * The two classes of FRBs discussed do not encompass the entire field. Although the authors introduce an additional class of FRBs, this does **not** address my question. Are there other classes? The authors should provide a more substantial argument rather than just listing additional classes and numerical results.
>
> Moreover, I believe other reviewers' comments also support my viewpoint. I have decided to maintain my current score.

---

### Author Response · Authors · 2025-11-24
**Summary of Our Rebuttal**

Dear Area Chairs and Reviewers,

We sincerely thank all reviewers for their thoughtful and constructive feedback. We are very encouraged that you found our work timely, well-motivated, and empirically valuable for understanding a critical failure mode in modern reasoning models. Your comments have directly helped us significantly strengthen the mechanistic clarity, methodological rigor, and empirical scope of the paper. Below we summarize the key strengths highlighted by reviewers and the concerns we fully addressed in our revisions.


## Key Strengths Highlighted by Reviewers

- **Importance and novelty of the problem:** Reviewers agreed that studying whether models over-trust reasoning-like structures is an important and well-posed direction, particularly as LRMs become widely adopted (Reviewers bZWX, X3gf).

- **Systematic and comprehensive evaluation:** The breadth of our empirical analysis—17 models across subjective and factual domains—was recognized as a major strength (Reviewers bZWX, YXr9).

- **Clear benchmarking design:** Reviewers appreciated the distinction between Simple Cues and Fake CoT, and noted that THEATER provides a concrete and extensible starting point for FRB evaluation (Reviewers X3gf, YXr9).

- **Well-organized and valuable findings:** Multiple reviewers found the results insightful, especially the identification of the “more thinking, less robust” paradox in LRMs (Reviewers bZWX, YXr9).


## Addressing Key Reviewer Concerns

**1. Need for deeper theoretical explanations (Reviewer bZWX)**
We added two new mechanistic experiments—**Pseudo-Formalism** and **Redundancy-Control**—and integrated them with a unified mechanistic explanation (reasoning-trace hijacking). These additions strengthen Section 3 and appendices, clarifying *why* FRB emerges rather than only *documenting* it.

**2. Distinguishing FRB from trivial redundancy or surface bias (Reviewers bZWX, aYJo, X3gf)**
We added:
- A **Neutral Control Cue** experiment showing that redundancy causes only noise-level fluctuation, while FRB cues cause systematic degradation.
- A **position-randomized attack** showing FRB persists regardless of option placement.
These results demonstrate that FRB is a semantic vulnerability, not a surface-level artifact.

**3. Statistical grounding and stability (Reviewer aYJo)**
We re-evaluated core experiments using **three random seeds**, added **paired and independent t-tests**, and included a **fine-grained temperature sweep (T = 0.0–1.0)**. Results show that FRB is statistically significant, seed-stable, and temperature-invariant.

**4. Generator bias in Fake CoT (Reviewers X3gf, YXr9)**
We added a **multi-generator ablation** (Claude, Gemini, Grok). FRB effects remained nearly identical, confirming generator-agnosticity.

**5. Asymmetry concerns & scaling to multiple-choice (Reviewer X3gf)**
We added a **Symmetric Dual-CoT setting**, where both options include reasoning. Accuracy nearly fully recovers, confirming FRB arises from asymmetric structural mimicry rather than formatting alone.

**6. Distinguishing internal vs. external reasoning effects (Reviewers X3gf, aYJo)**
We added:
- **CoT-suppression experiments**, showing that “no CoT” mitigates FRB on subjective tasks.
- Additional analysis explaining why **Shallow CoT is more damaging** than Deep CoT.
- Expanded discussion on why **LRMs outperform LLMs on factual Fake CoT**, revealing verification advantages.

**7. Broader novelty & relationship to prior work (Reviewer YXr9)**
We clarified distinctions between FRB and prior noise-fragility works.
- Prior studies primarily measure *distraction*.
- FRB reveals *assimilation* of fake reasoning and metacognitive distortion.
We added comparison tables and expanded the related work discussion.

**8. Ecological validity of the benchmark (Reviewers X3gf, YXr9)**
We explicitly connected our setup to real-world **LLM-as-a-Judge** pipelines, where models must evaluate fixed candidate responses containing reasoning-like structures.

**9. Mitigation robustness and CoT behavior after mitigation (Reviewer X3gf)**
We performed:
- A **Prompt Robustness Ablation** with multiple prompt variants (variance < 0.5%).
- A **Post-Mitigation Trace Analysis**, explaining why mitigation succeeds in factual tasks (“verification interrupt”) but fails or backfires on subjective tasks.


We sincerely thank all reviewers for their valuable feedback. We have substantially strengthened the paper through extensive new experiments, deeper mechanistic explanations, improved clarity, and expanded discussion. We hope these revisions adequately address all concerns and kindly ask you to consider raising your scores.

Best regards,

All Authors

---

> ### Comment · Reviewer_aYJo · 2025-11-26
> **Information overload**
>
> I feel that some kind of unspoken rule or etiquette is breached when there is a paper's worth of novel content thrown in response to each reviewer, which creates a lot of work in an asymmetric fashion. The volume of litigation in these reviews feels like it exceeds the conceptual contents of the paper by a lot; are these reviews LLM-generated or is this all genuine legwork? If it is all genuine legwork, why wasn't this much thought put into the original paper?
>
> I will try to make my way through the rebuttals, but all of them appear to follow a common template; walls-of-text followed by a request for a score raise. The contents may be fine, but this can be less charitably interpreted as a "workload intimidation" strategy where the easier course of action for all reviewers is to just raise score rather than read 4 new papers worth of content.

---

### Note · Authors · 2025-12-24

**Comment:**

We decided to withdraw this paper due to other plans. Thanks to ACs and reviewers for your work.

**Withdrawal Confirmation:**

I have read and agree with the venue's withdrawal policy on behalf of myself and my co-authors.